# Identification and characterization of CIM-1, a carbapenemase that adds to the family of resistance factors against last resort antibiotics
Yu Wang [1,2], Sylvia A. Sapula [1], Jonathan J. Whittall[1], Jack M. Blaikie[1], Olga Lomovskaya [3] & Henrietta Venter [1] ✉

The increasing rate of carbapenem-resistant bacteria within healthcare environments is an issue of great concern that needs urgent attention. This resistance is driven by metallo-β-lactamases (MBLs), which can catalyse the hydrolysis of almost all clinically available β-lactams and are resistant to all the clinically utilized β-lactamase inhibitors. In this study, an uncharacterized MBL is identified in a multidrug resistant isolate of the opportunistic pathogen, *Chryseobacterium indologenes*. Sequence analysis predicts this MBL (CIM-1) to be a lipoprotein with an atypical lipobox. Characterization of CIM-1 reveals it to be a high-affinity carbapenemase with a broad spectrum of activity that includes all cephalosporins and carbapenems. Results also shown that CIM-1 is potentially a membrane-associated MBL with an uncharacterized lipobox. Using prediction tools, we also identify more potentially lipidated MBLs with non-canonical lipoboxes highlighting the necessity of further investigation of lipidated MBLs.

The development of antimicrobial resistance (AMR) poses a major threat to human health and represents a growing burden to the global healthcare system. In 2019, 4.95 million deaths were associated with bacterial AMR worldwide[1]. It has also been estimated that AMR will surpass cancer as the leading cause of death by 2050 and that 10 million people are estimated to die each year if policies remain unchanged[2]. Among all the current treatments, β-lactams are the most frequently prescribed antibiotics[3]. Unfortunately, resistance to β-lactams is widespread and increasing expeditiously worldwide[1]. Resistance to β-lactams is associated with poor clinical outcomes and a high economic burden, with longer hospital stays contributing to increased hospitalisation costs[4]. Carbapenems are β-lactam antibiotics that are a last-resort treatment for multidrug-resistant bacterial infections[5]. The increasing rate of carbapenem resistance, especially in opportunistic pathogens that are intrinsically resistant to colistin, such as *Chryseobacterium indologenes*, severely limits the therapeutic options and represents a major public health concern.

*Chryseobacterium* spp. are multidrug-resistant bacterial species which are known to cause hospital-acquired infections in immunocompromised patients. Among the six species under this genus, the incidence of *C.*

*indologenes* infections has been increasing for the past two decades[6,7]. These rod-shaped, yellow-pigmented, Gram-negative bacteria are ubiquitous in environmental soil and water. They can also be found in hospital environment on wet, humid surfaces such as respirators, feeding tubes, and indwelling devices[8]. Notably, the immunocompetent population can also be infected by *C. indologenes*, thus these organisms are not only problematic in hospital settings but also in the community environment[9–11]. Additionally, *C. indologenes* is frequently isolated from cystic fibrosis patients with other opportunistic pathogens, including *Pseudomonas aeruginosa* and *Burkholderia cepacia*[12]. In a recent study, *Chryseobacterium* was also found to be one of the most prevalent genera isolated from the lungs of COVID-19 patients[13].

The production of β-lactamases, enzymes that can hydrolyse and inactive β-lactams, is the most common resistance mechanism utilised by Gram-negative bacteria against β-lactam antibiotics[14]. β-lactamases can be broadly classified into two classes based on their catalytic centre: the serine-β-lactamases and the metallo-β-lactamases (MBLs). The latter are more concerning due to their ability to hydrolyse all bicyclic β-lactams, including the last resort antibiotics, carbapenems[15]. Carbapenem-resistant organisms

[1]Health and Biomedical Innovation, Clinical and Health Sciences, University of South Australia, Adelaide, Australia. [2]School of Biomedical Science, University of Adelaide, Adelaide, Australia. [3]Qpex Biopharma, Inc., San Diego, CA, USA. ✉e-mail: rietie.venter@unisa.edu.au

were listed as the top category in the World Health Organisation's critical priority of most dangerous pathogens in urgent need of new antibiotic development[16]. To overcome this resistance mechanism, β-lactamase inhibitors such as clavulanic acid and the latest, vaborbactam, were developed to restore or preserve the efficacy of β-lactams[16]. Despite decades of effort, none of the marketed β-lactamase inhibitors have been found to be active against MBLs, making MBLs a global concern. However, there are some β-lactamase inhibitors under development that show preliminary effects against MBLs[17–19]. Notably, xeruborbactam, apart from its broad spectrum against both serine and metallo-β-lactamase, was shown to have direct antibacterial activity in some organisms due to its inhibition of penicillin-binding proteins[20].

MBLs can further be classified into three subclasses (B1–B3) based on amino acid sequence identity and zinc ion dependence. Subclass B1 MBLs have the broadest substrate profile, including both cephalosporins and carbapenems. Of these, the New Delhi Metallo-β-Lactamase (NDM) is the most well-characterised and an example of the most disseminated MBL[21]. Since the identification of this enzyme, NDM-producing bacteria have been reported worldwide, and NDM variants have become the most clinically relevant MBLs[22]. NMD is a lipoprotein with a highly conserved lipobox sequence (LSGC) and is tethered to the outer membrane[23]. Additionally, the lipidation of NDM potentially prevents it from degradation under Zn(II) starvation condition and aids secretion of this enzyme in outer membrane vesicles (OMVs)[23]. Although 43 other MBLs were predicted as lipoproteins in the study performed by Gonzalez et al.[23] using the lipobox conserved sequence, NDM remains the only characterised MBL to date.

Bacterial lipoproteins can be identified using the consensus sequence of the lipobox located at the C region of the signal peptide. The conserved sequence has expanded from [L][AS][GA][C] based on 26 distinct lipoprotein precursors to [LVI][ASTVI][GAS][C] through the decades as more and more lipoproteins have been identified[24–26]. Due to the advances in computer technology, several prediction tools were developed, including SignalP 6.0, which uses a machine learning model to detect all known types of signal peptides[27].

Here we report the identification of an uncharacterised subclass B1 MBL from the opportunistic pathogen, *C. indologenes*, termed CIM-1 (*C. indologenes* MBL). The resistance profiles of CIM-1, together with NDM-4, and a periplasmic MBL from *C. indologenes*, IND-2, were investigated in *Escherichia coli* C41(DE3). Upon purification of these MBLs, kinetic activities on various β-lactams were investigated. Furthermore, the incidence of potential lipidation in MBLs was investigated for CIM-1 homologues.

## Results

### Origin of *C. indologenes* #3362

*C. indologenes* #3362 was isolated in 2019 from a sink swab sample from a residential aged care facility as part of an antimicrobial surveillance project. The genome size of *C. indologenes* is 5105493 bp with 37.28% GC content (Accession number: JAPSGE000000000). Preliminary antimicrobial susceptibility assays indicated high-level intrinsic resistance to ceftazidime, imipenem, meropenem and colistin. *C. indologenes* is an opportunistic pathogen that is intrinsically resistant against the last resort antibiotics carbapenems and colistin. Hence the presence of this pathogen in close proximity of a vulnerable population warranted a closer investigation into its resistance profile and genomic content.

### Bioinformatic analysis of *C. indologenes* isolate #3362 revealed a putative subclass B1 β-lactamase

Whole genome sequencing was performed on a resistant *C. indologenes* isolate #3362. Sequencing analysis revealed three β-lactamases including an Ambler Class A Extended-Spectrum β-lactamase, *bla*CIA, and two MBLs, *bla*IND-2 and a putative MBL gene. The *bla*CIA and *bla*IND-2 genes code for two predominant β-lactamase genes in *C. indologenes* and co-harbouring of both has been reported worldwide[28–30]. The putative MBL is composed of 251 amino acids and has not been characterised before

hence was named CIM-1 for *C. indologenes* metallo-β-lactamase 1 in this study.

CIM-1 was further assessed by an amino acid sequence alignment with the most widely disseminated MBL, the New-Dehli metallo β-lactamase. CIM-1 and NDM-1 revealed low sequence identity (33.5%). However, some level of similarity was observed in the region of loops 3 and 10, which have been identified as important for substrate binding[31,32] (Fig. 1a). Assessment of known conserved regions found within MBLs revealed that CIM-1 has all the highly conserved residues in the zinc-binding site of class B1 MBLs (Fig. 1a).

The primary amino acid sequence of CIM-1 was used to generate a 3D-CIM-1 model using the SWISS-MODEL server (Fig. 1b), with results identifying an MBL carbapenemase, MYO-1 (PDB: 6T5L) as the top template. MYO-1 is a plasmid-encoded subclass B1 MBL isolated from *Myroides odoratimimus*. MYO-1 and CIM-1 were predicted to share 49% amino acid sequence identity (Supplementary Fig. 1). The model generated using this template was shown to be of high quality and high reliability, displaying a high GMQE (global model quality estimate)[33] score of 0.81 and high QMEANDisCo (distance constraints applied on model quality estimation)[34] global score of 0.84. The active site of CIM-1 hosts two Zn (II) ions in a shallow grove in between loop 3 and 10 (Fig. 1b). The confidence level of the model at loop L3 region is relatively low despite the high amino acid sequence similarity (Fig. 1b). This may be due to the flexibility of this loop as it is involved in substrate binding. The highly conserved Zn1 coordinators (His116, His118, and His196) and Zn2 site coordinators (Asp120, Cys221, and His263) among subclass B1 MBLs are presented in the CIM-1 amino acid sequence as denoted in Fig. 1a and presented in Fig. 1c.

### N-terminal sequences alter protein localisation, expression level, and resistance profiles of MBL in *E. coli*

As MBLs are typically translocated into the periplasmic space, the CIM-1 signal peptide and translocation pathway were assessed by SignalP 6.0[27]. CIM-1 was predicted to be a lipoprotein transported by the Sec-translocation pathway and cleaved by signal peptidase II at a high likelihood (0.9909, Supplementary Fig. 2). The cleavage site of the signal peptide was predicted between Asn 18 and Cys 19. During lipoprotein biosynthesis, signal peptide sequences are cleaved, and Cys become the first amino acid of the mature protein sequence (Fig. 2a). The lipobox sequence, which is a four amino acid sequence including the three in front of the Cys, is important for this lipoprotein biosynthesis and is used for prediction of these proteins. Interestingly, assessment of the lipobox revealed that CIM-1 has a non-canonical lipobox in comparison to the canonical lipobox exemplified by NDM (Fig. 2b). The only conserved amino acid within the CIM-1 sequence is Cys (C19), which is essential in the attachment of lipid moiety during lipoprotein biosynthesis[26]. To explore whether removing Cys would disrupt the membrane association of CIM-1, the CIM-1 C19A mutant was generated. For a comparison, the periplasmic MBL, IND-2 from *C. indologenes* #3362 and NDM-4 from *Klebsiella pneumoniae* D53 was also cloned and expressed in *E. coli* C41(DE3).

To further investigate the localisation of CIM-1 and the effect of the N-terminal amino acid sequence on protein localisation, signal peptide exchange constructs were made (Fig. 2c). I-CIM and C-IND were constructed to assess if the exchange of signal peptides and the first three amino acids of the mature protein will be able to alter protein localisation (Fig. 2c). As shown in Fig. 2c, the exchanged region includes signal peptides till +3 position after the signal peptide cleavage site as a few amino acids after Cys can affect protein sorting by the Lol system[35]. Asp at the +2 position can lead to inner membrane retention, and some amino acids, including Asp, Glu, and Gln at the +3 position, can strengthen the retention and thus are referred to as the Lol avoidance signal[35]. This avoidance signal was not identified in the CIM-1 amino acid sequence, suggesting that CIM-1 may be able to be translocated to the outer membrane via the Lol pathway, so we speculate that it may be located in the inner leaflet of the outer membrane (Fig. 2d). As such, based on their signal peptides and the +2 amino acid, we postulated the cellular localisation of the constructs (Fig. 3a). As NDM is the

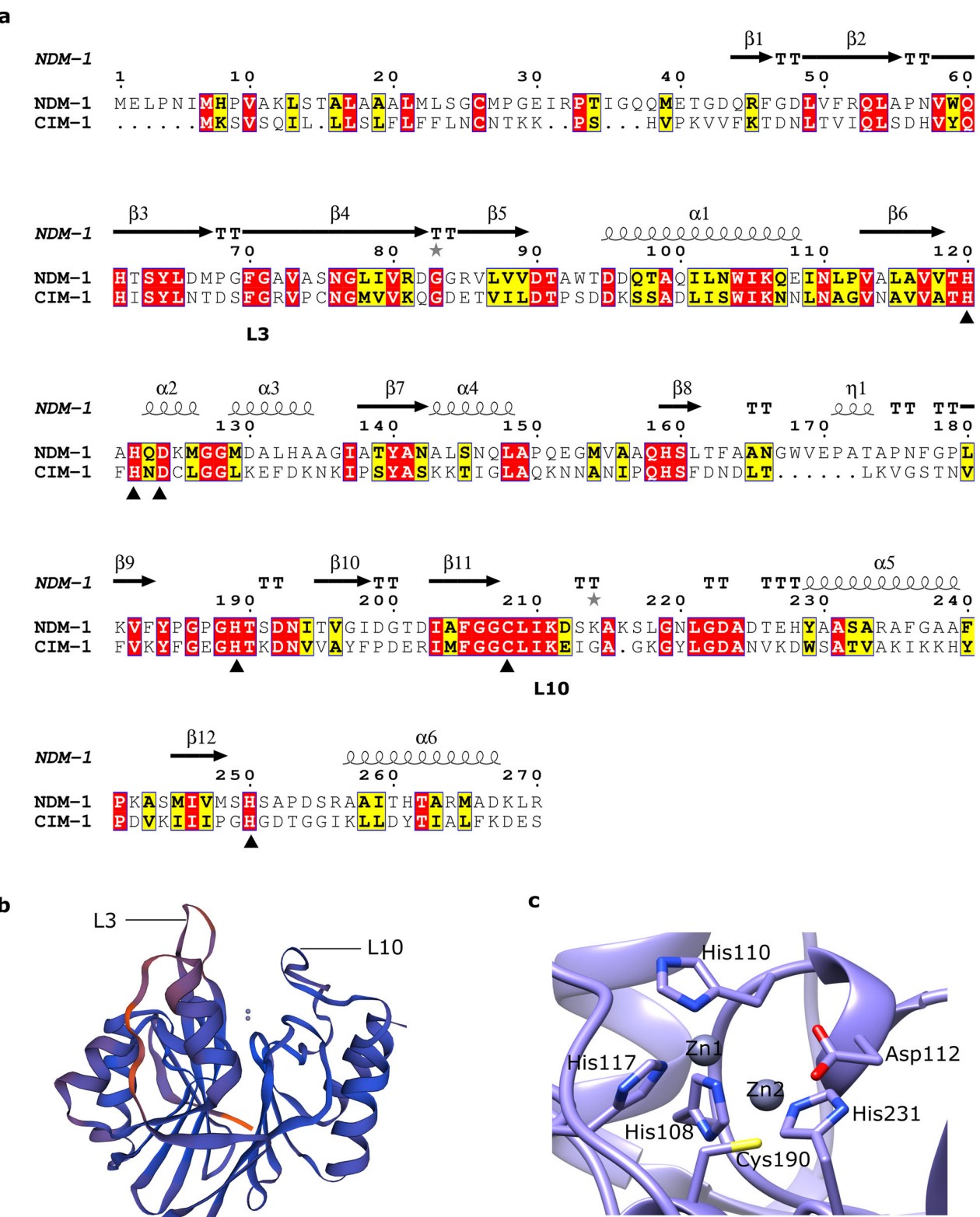

**Fig. 1 | Identification of an uncharacterised metallo-β-lactamase in *C. indologenes*. a** Amino acid sequence alignment of CIM-1 (accession number: WP_123865086.1) with NDM-1 generated by ESPript 3.0. The published structure of NDM-1 (PDB ID: 3ZR9) was used for calculating the secondary structure elements. Sequence identity between NDM-1 and CIM-1 was determined as 33.5%. Zinc coordinators are denoted in black triangles. **b** 3D model of CIM-1 generated based on MYO-1 (PDB: 6T5L) the MBL from *M. odoratimimus*. Blue residues indicate a high level of confidence in the generated model. Red residues indicate a low level of confidence. **c** Zinc binding residues in CIM-1 3D structure generated by SWISS-MODEL.

**Fig. 2 | Schematic representation of constructs and analysis of important amino acids. a** Biosynthesis of lipoproteins. (1) attachment of diacylglyceryl moiety to the thiol group of Cys via thioether linkage, (2) cleavage of the signal peptide, (3) attachment of acylglyceryl to the free amino group of Cys. Red: lipobox, orange: N-terminal sequence of the signal peptide. **b** Comparison of the lipobox sequences of CIM-1 and NDM-1. **c** Schematic representation of the constructs designed and assessed in this study. **d** '+2' rule and the amino acids after the highly conserved Cys in CIM-1 and NDM-4.

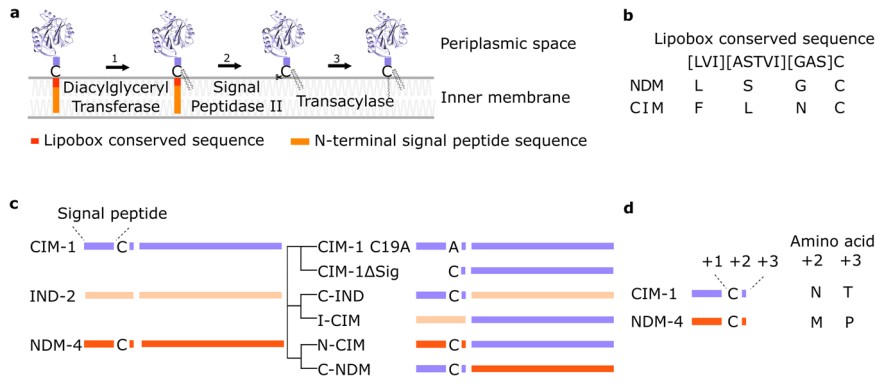

only characterised lipidated MBL to date, N-CIM and C-NDM were constructed to assess whether exchange of signal peptides between two lipidated proteins will have any effect on protein localisation and their resistance profiles. Finally, the mature protein of CIM-1 with the signal peptide removed (ΔSig) was constructed to generate a soluble version of CIM-1. As shown in Table 1, all the constructs are viable and able to confer resistance to *E. coli*.

As shown in Fig. 3b, CIM-1 and NDM-4 were indeed detected in the membrane fractions, while IND-2 was detected in both membrane and soluble fractions (Fig. 3b). Two bands of different sizes can be observed in membrane fractions of I-CIM and N-CIM. The larger band is believed to represent the precursor protein (full-length), which could be contamination from the cytoplasmic precursor protein or un-cleaved protein which has transient interactions with membranes. The smaller-sized protein observed in the membrane and soluble fractions is deemed to be the mature (folded) protein. The interaction between IND-2 and membrane is potentially caused by the electrostatic interactions through a positively charged protein surface of IND-2 (Supplementary Fig. 3). Results also revealed that replacing the periplasmic signal peptide with a lipoprotein signal peptide leads to protein relocation as C-IND was found to be localised to the membrane fraction. Interestingly, replacing the lipoprotein signal peptide with a periplasmic signal peptide did not fully alter the protein localisation as I-CIM was identified in both membrane and soluble fractions. However, the majority of the mature protein was localised in the soluble fraction, suggesting that the exchange of CIM-1 and IND-2 N-terminal sequences did alter the protein localisation. Lastly, removing the N-terminal sequence produced a soluble version of CIM-1.

The resistance profiles of the various MBL constructs were determined for plasmids transformed into *E. coli* C41(DE3) cells. CIM-1 was able to confer resistance upon *E. coli* to all the β-lactam antibiotics tested apart from cefepime, albeit at lower levels (lower MIC values) compared to IND-2 and NDM-4. However, CIM-1 is clearly able to hydrolyse cefepime, as is indicated by the high MIC value that CIM-1 ΔSig confers to the *E. coli* cells. The differences observed in resistance profiles directly correlate with the expression levels of the various constructs (Supplementary Fig. 4).

The general cephalosporin resistance trend revealed that all constructs had reduced β-lactamase activity against the later-generation cephalosporins (Table 1). As expected, none of the constructs conferred resistance to the monocyclic β-lactam (aztreonam). The unusually high resistance profile observed for CIM-1ΔSig expressing *E coli* cells was unexpected. This construct should be in the cytoplasm as its signal peptide, essential for translocation to the periplasm, was removed, and the site of action of β-lactam antibiotics is in the periplasm. We postulate that this high resistance was due to the cytoplasmic CIM-1ΔSig not being able to reach the periplasmic space where the β-lactam attacks, leading to cell lysis and the release of cell content. High expression levels of CIM-1ΔSig (Supplementary Fig. 5) and its presence within the medium may thus be deactivating antibiotics in the broth allowing the remaining cells to grow in this antibiotic-free environment. This result suggests that CIM-1 is a very active enzyme against all tested bicyclic β-lactams.

## CIM-1 is potentially a membrane-tethered MBL
CIM-1 was predicted to be a membrane-tethered protein, and the protein localisation assay revealed that CIM-1 was mainly observed in *E. coli* C41(DE3) membrane fraction, indicating that it is potentially a membrane-associated protein (Fig. 3b). Cysteine is critical in lipidation as it serves as a linkage of lipoprotein and membrane through attachment of a lipid moiety (Fig. 3a). Replacing Cys with Ala were expected to produce a soluble version of CIM-1. However, this mutation did not abolish the membrane localisation of CIM-1 (Fig. 3b). In order to identify the lipid-modified peptide sequence of the mature CIM-1 protein, CIM-1 was subjected to liquid chromatography-tandem mass spectrometry (LC–MS/MS). Almost all peptides of the mature CIM-1 protein were identified (Supplementary Fig. 6). The N-terminal peptide sequence of mature CIM-1 with or without triacylated Cys was not observed. The detection of lipidated species can be very challenging in this case due to the many possibilities of modified groups to the lipidated Cys[36].

To better investigate protein localisation in its native host, the membrane fraction was isolated from *C. indologenes* #3362 and analysed by LC-MS/MS. Both CIM-1 and IND-2 can be identified from the membrane fraction (Supplementary Fig. 7). Relative protein abundance in the membrane fraction was normalised by protein expression level measured by quantitative reverse transcription polymerase chain reaction (RT-qPCR) (Table 2, Supplementary Table 1). The amount of CIM-1 in membrane fraction is 14.88-fold compared to IND-2. It suggested that CIM-1 is potentially a membrane-associated protein.

CIM-1 was predicted to be tethered to the outer membrane, which also means that it will be targeted to the OMVs. To assess this, OMVs were isolated from a culture supernatant of *C. indologenes* and examined under transmission electron microscopy (TEM). Spherical vesicles with a diameter between 50 to 200 nm were observed (Supplementary Fig. 8). Both CIM-1 and IND-2 were identified from the isolated OMVs using LC–MS/MS. The ratio of protein abundance in membrane and OMV also supported that CIM-1 is potentially a membrane tethered protein (Table 2).

## CIM-1 has a high affinity for cephalosporins and carbapenems
Kinetic assays were performed on purified proteins to compare the biochemical properties of the three enzymes, including their abilities to hydrolyse various classes of β-lactam antibiotics. Hydrolysis in the presence of the MBL inhibitor 1,10-phenanthroline (zinc chelating agent) was included as a control for meropenem. 1,10-phenanthroline was able to inhibit 70% of the hydrolysis activity (Supplementary Fig. 9).

All three enzymes (IND-2ΔSig, CIM-1ΔSig, and NDM-4ΔSig) showed catalytic activity against the substrates tested in this study, with the exception of IND-2, which was unable to hydrolyse cefepime (Table 3). The two MBLs carried by *C. indologenes* have very different kinetic properties. CIM-1ΔSig was broadly active against all seven β-lactam substrates in the study, showing the highest affinity for nitrocefin, ertapenem and imipenem with $Km$ values less than 100 µM and relatively high turnover rates against ceftazidime. IND-2ΔSig had high $Km$ values (low affinity) against all tested substrates apart from nitrocefin, for which it had a $Km$ at 115 µM. These in

**a**

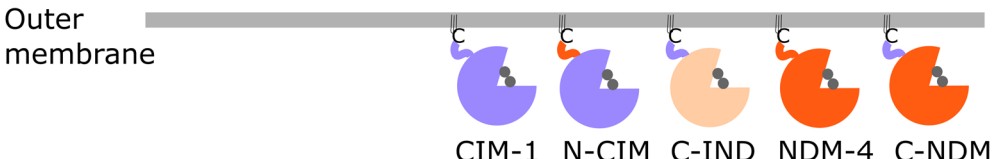

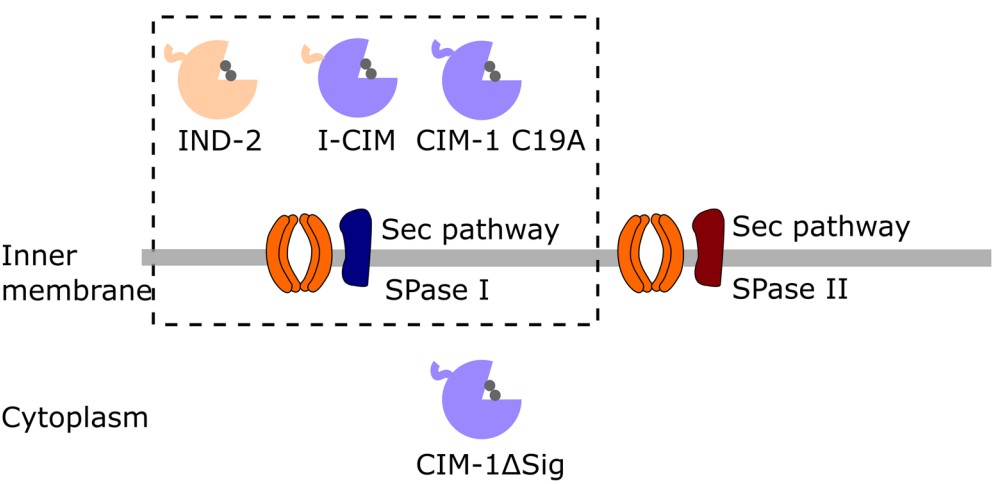

**b**

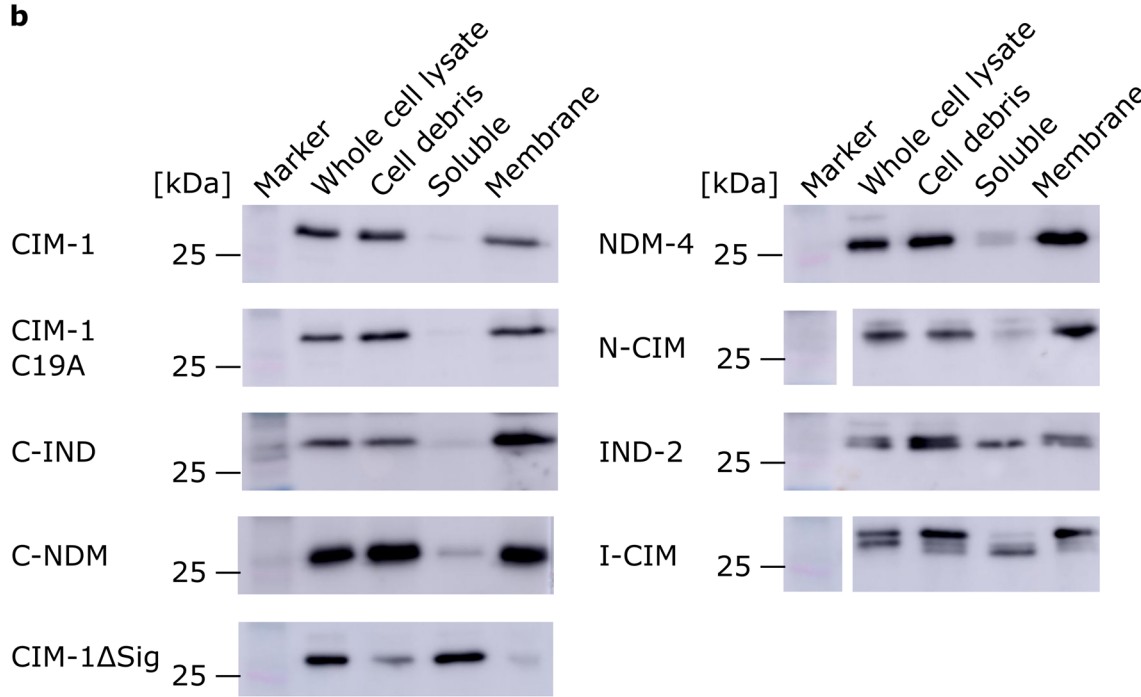

**Fig. 3 | Schematic representation of constructs and their postulated cellular localisation. a** Schematic representation of the postulated cellular localisation of the constructs. **b** Representative Western blot images of protein levels of various protein constructs in whole cells and cellular fractions from *E. coli* C41(DE3). THE™ His tag monoclonal mouse antibodies and Invitrogen™ Goat anti-Mouse Immunoglobulin G were used in immunodetection. Blot images of IND-2 and N-CIM were derived from the same membrane, thus sharing the same protein marker. CIM-1ΔSig and I-CIM were also derived from the same membrane. A white space for blot images of N-CIM and I-CIM was left in between the strips as the markers, and the samples are not from adjacent lanes.

Table 1 | Minimum inhibitory concentrations (MIC) for *E. coli* C41(DE3) expressing CIM-1, IND-2, NDM-4, the Cys mutant CIM-1 C19A, the chimeric constructs I-CIM, C-IND, C-NDM, N-CIM, and the construct lacking the signal peptide CIM-1ΔSig sub-cloned into the pET41a(+) vector

| β-lactams | MIC (µg/mL) for *E. coli* C41(DE3) (fold-change relative to MIC of pET41a(+)) | | | | | | | | | | |
|---|---|---|---|---|---|---|---|---|---|---|---|
| | *C. indologenes* #3362 | pET41a(+) | CIM-1 | CIM-1 C19A | NDM | IND-2 | I-CIM | C-IND | N-CIM | C-NDM | CIM-1ΔSig |
| Ampicillin | 512 | 4 | 8 (2) | 8 (2) | >128 (>32) | >128 (>32) | >128 (>32) | 64 (16) | 64 (16) | 256 (64) | >128 (>32) |
| *Cefazolin (1st) | 256 | 2 | 4 (2) | 4 (2) | >128 (>64) | 64 (32) | 128 (64) | 2 (1) | 64 (32) | 64 (32) | 128 (64) |
| *Cefoxitin (2nd) | 32 | 0.5 | 1 (2) | 1 (2) | >128 (>256) | 32 (64) | 32 (64) | 1 (2) | 16 (32) | 16 (32) | >128 (>256) |
| *Ceftazidime (3rd) | 16 | 0.125 | 0.25 (2) | 0.25 (2) | >128 (>1024) | 0.25 (2) | 64 (512) | 0.125 (1) | 64 (512) | 32 (256) | >128 (>1024) |
| *Cefepime (4th) | 2 | 0.031 | 0.031 (1) | 0.031 (1) | 64 (2048) | 0.031 (1) | 16 (512) | 0.031 (1) | 8 (256) | 0.5 (16) | >128 (>4096) |
| Doripenem | 128 | 0.063 | 0.25 (4) | 0.5 (8) | >128 (>2048) | >128 (>2048) | 128 (2048) | 0.25 (4) | 16 (256) | 8 (128) | >128 (>2048) |
| Ertapenem | 64 | 0.016 | 0.063 (4) | 0.063 (4) | 64 (4000) | 128 (8000) | 16 (1000) | 0.016 (1) | 4 (256) | 2 (128) | >128 (>8000) |
| Imipenem | 128 | 0.5 | 2 (4) | 2 (4) | >128 (>256) | >128 (>256) | 128 (256) | 2 (4) | 16 (32) | 8 (16) | >128 (>256) |
| Meropenem | 128 | 0.031 | 0.125 (4) | 0.25 (8) | >128 (>4096) | >128 (>4096) | 128 (4096) | 0.063 (2) | 16 (512) | 8 (256) | >128 (>4096) |
| Aztreonam | >512 | <0.02 | <0.02 | <0.02 | <0.02 | <0.02 | <0.02 | <0.02 | <0.02 | <0.02 | <0.02 |

*E. coli* C41(DE3) propagating the pET41a(+) vector was used as a control. *C. indologenes* #3362, carrying both IND-2 and CIM-1, was used as a comparator.
*The generation of cephalosporins is indicated in brackets for each cephalosporin.

## Table 2 | CIM-1 is identified in the membrane fraction of *C. indologenes*

| Protein | Relative abundance in membrane fraction | Ratio (Membrane/OMV) |
|---|---|---|
| CIM-1 | 14.88 | 2.19 |
| IND-2 | 1 | 0.528 |

Relative protein abundance of CIM-1 and IND-2 presented in the membrane fraction of C. indologenes #3362 normalised by protein expression level.

vitro results suggest that CIM-1 is likely to be an effective and clinically important broad-spectrum MBL. Comparing CIM-1 with NDM-4, which is the most disseminated and clinically relevant MBL, similar low $Km$ values were observed, especially against carbapenems. However, compared to IND-2, a lower turnover rate was observed for CIM-1 which was comparable with the turnover rates of NDM-4. In addition, CIM-1 was observed to have a lower affinity for cephalosporin than NDM-4.

### Homologues of CIM-1 distributed among environmental bacteria are predicted to be lipoproteins

To investigate whether CIM-1 is carried by bacteria other than *C. indologenes*, a BlastP search was performed with the amino acid sequence of CIM-1. Homologues of CIM-1 were found distributed widely across various species of mainly environmental bacteria (Fig. 4). The sequence identity of between the top 100 homologues and CIM-1 is between 100% (Accession number: WP_123865086.1) and 54% (Accession number: WP_035589998.1). A phylogenetic tree comprised of the top 100 CIM-1 homologues revealed that there are three major clusters (Fig. 4). Of these, clade 1 contained the smallest group of four proteins from *Hymenobacter* spp., Gram-negative bacteria found in soil and water[37,38]. Clade 2 contained a larger number of proteins from a diverse group of bacterial species. Assessment of the lipoboxes revealed that the two homologues from *Flavobacterium jejuense* and one from *Avrilella dinanensis* were predicated to be lipoproteins with a high likelihood that fulfils the conserved lipobox sequence ([LVI][ASTVI][GAS]C), while the rest of the predicted lipoproteins did not follow the conserved sequence. The second clade contained a broad variety of environmental bacteria, mainly found in water and soil. Sequence alignments show that two protein homologues from *Pedobacter* (Accession number: RZK54418.1 and RZL36466.1) and two from *Elizabethkingia anopheles* (Accession number: WP_035589998.1 and MBG0505245.1) carry a highly conserved mature protein sequence, but a low similarity in signal peptide sequence (Supplementary Fig. 10). The third clade contained mostly the *Chryseobacterium spp.* homologues, with most predicted to be lipoproteins with a variety of lipobox sequences. We also observed that lipobox sequences are well-conserved within bacterial species, although assessment of this atypical lipobox requires further study.

### Discussion

In this study, a putative MBL (CIM-1) was identified in a *C. indologenes* isolate recovered from a healthcare facility. *C. indologenes* is the most frequently isolated of the *Chryseobacterium* species and is intrinsically resistant to several classes of antibiotics, including penicillin, cephalosporins, and the last resort antibiotics, carbapenems and colistin. This *C. indologenes* also carries another MBL, IND-2. Both IND-2 and CIM-1 are subclass B1 MBLs which have a broader substrate profile, including all the bicyclic β-lactams, compared to subclass B2 (carbapenems) and B3 (penicillins and cephalosporins) MBLs. IND-2 was first isolated from *C. indologenes* in 2000[39]. However, CIM-1 was never identified or characterised and shares only a low sequence identity with IND-2. Hence, CIM-1, as an uncharacterised subclass B1 MBL, was the focus of this study.

Our analysis predicted IND to be a secretory protein which goes through the general secretory (Sec) pathway and is cleaved by signal peptidase I (SPase I) like many other MBLs[40]. Our results show that IND-2 has some association with the membrane in both *E. coli* and the native host *C. indologenes*.

**Table 3 | Kinetic parameters of recombinantly expressed and purified CIM-1ΔSig, IND-2ΔSig, and NDM-4ΔSig**

| β-lactams substrates | $K_m$ (µM) | | | $k_{cat}$ (s$^{-1}$) | | | $k_{cat}/K_m$ (s$^{-1}$ M$^{-1}$) | | |
|---|---|---|---|---|---|---|---|---|---|
| | IND-2ΔSig | CIM-1ΔSig | NDM-4ΔSig | IND-2ΔSig | CIM-1ΔSig | NDM-4ΔSig | IND-2ΔSig | CIM-1ΔSig | NDM-4ΔSig |
| Nitrocefin | 113 ± 6.9 | 33 ± 6 | 11 ± 1.8 | 78 | 2.8 | 3.4 | $6.8 \times 10^5$ | $8.5 \times 10^4$ | $3.1 \times 10^5$ |
| Ceftazidime | >1000 | 324 ± 42 | 93 ± 14 | ND | 30 | 6.3 | ND | $9.3 \times 10^4$ | $6.8 \times 10^4$ |
| Cefepime | NH | 538 ± 56 | 136 ± 16 | – | 15.3 | 5.4 | – | $2.8 \times 10^4$ | $4.0 \times 10^4$ |
| Doripenem | >1000 | 175 ± 24 | 253 ± 19 | ND | 21 | 51 | ND | $1.2 \times 10^5$ | $2.0 \times 10^5$ |
| Ertapenem | 1069 ± 105 | 91 ± 11 | 159 ± 22 | 127 | 9.2 | 17 | $1.2 \times 10^5$ | $1.0 \times 10^5$ | $1.1 \times 10^5$ |
| Imipenem | 728 ± 102 | 94 ± 7 | 151 ± 22 | 208 | 8.5 | 31 | $2.9 \times 10^5$ | $9.0 \times 10^4$ | $2.1 \times 10^5$ |
| Meropenem | >1000 | 143 ± 19 | 163 ± 22 | ND | 15 | 25 | ND | $1.0 \times 10^5$ | $1.5 \times 10^5$ |
| Aztreonam | NH | NH | NH | – | – | – | – | – | – |

*NH* no hydrolysis was detected, *ND* not determined.
Data shown are the results generated using Prism GraphPad 8.0 from three independent replicates. Errors are reported as standard errors.

Furthermore, IND-2 was identified from the OMVs isolated from *C. indologenes*. This is not unexpected as soluble MBLs can be secreted through OMVs by transient electrostatic interactions[41].

CIM-1 was predicted to be a lipoprotein and cleaved by signal peptidase II at high likelihood (0.9907), with the cleavage site postulated to be localised at the highly conserved Cys (C19). Research into bacterial lipoproteins has identified a highly conserved lipobox sequence, [LVI][ASTVI][GAS]C, as exemplified by NDM-1[26]. However, CIM-1 was observed to carry a non-canonical lipobox, FLNC, with the only conserved amino acid being the Cys, which is vital in the attachment of lipid moiety and lipoprotein biosynthesis[26]. This non-canonical lipobox is not totally impossible, Phe(F) and Leu(L) was observed at position -3 and -2 at a lower frequency and thus were not included in the consensus sequence[25]. In our study, we found that the protein abundance of CIM-1 presented in membrane fraction is significantly higher than IND-2, as well as a higher ratio of membrane/OMVs, indicating that CIM-1 is likely a membrane-associated protein.

With the expansion of the lipoprotein database, degeneration of the lipobox conserved sequence was observed especially within spirochaetes[42,43]. Is it a simple degeneration of the lipobox sequence or could variant lipobox sequences indicate its functional differences? These are questions to be answered to better understand the lipidated β-lactamases.

Investigation of the antimicrobial susceptibility profile of recombinant MBLs expressed in *E. coli* revealed that all the constructs are viable and able to confer resistance to *E. coli*. CIM-1 expressed in *E. coli* conferred low levels of resistance (2- to 4-fold increase in MIC) against almost all β-lactams tested in this study. Compared with CIM-1, IND-2 confers higher levels of resistance, especially towards carbapenems. MIC variations were also observed among chimeric MBLs, which were postulated to result from different expression levels (Supplementary Fig. 4). While the amount of MBLs expressed matters for the observed phenotypic variation, other factors such as quantity of mRNA and the efficacy of enzymes are also important[44]. Low expression levels of constructs carrying the CIM-1 signal peptide, including CIM-1, CIM-1 C19A, C-IND, and C-NDM suggest that the lipoprotein signal peptide native to *C. indologenes* may not be compatible with the expression system of *E. coli*. The kinetic assays performed on purified CIM-1ΔSig show that CIM-1 has a much higher affinity against all tested cephalosporins and carbapenems compared to IND-2. Additionally, CIM-1 is active against the fourth-generation cephalosporin cefepime, which IND-2 is not able to hydrolyse. The comparable kinetic properties of NDM-1 and CIM-1 suggest that lipidated MBLs might have a higher affinity for β-lactams than periplasmic MBLs. This would be of advantage when they are excreted in OMVs, into the surrounding media where the concentration of antibiotics is significantly lower than in the periplasmic space. A high affinity for a broad spectrum of antibiotics allows MBLs to hydrolyse low levels of antibiotics in the environment and to protect antibiotic susceptible bacteria in the surroundings. As shown in other studies, many β-lactamases that are secreted within OMVs can protect antibiotic susceptible bacteria in their surroundings against these antibiotics[23,45]. This phenomenon has not only been observed experimentally but has now been reported clinically[46,47]. In our study, we identified both CIM-1 and IND-2 in the secreted OMVs from *C. indologenes*, demonstrating their ability to protect surrounding antibiotic susceptible bacteria. As CIM-1 has a high affinity for cephalosporins and carbapenems, it will be very effective in deactivating these antibiotics to provide resistance to an entire mixed microbial population when excreted in OMVs. CIM-1 may play a similar role in protecting antibiotic susceptible bacteria, which is somewhat concerning as *C. indologenes* is frequently isolated from cystic fibrosis and COVID-19 patients[12,13], and may hamper antibiotic treatment.

The identification of CIM-1 as an uncharacterised carbapenemase motivated us to search for CIM-1 homologues. Most of the homologues were from the *Flavobacteriales* order of environmental organisms. Some of these are opportunistic pathogens that have been shown to cause diseases, including *C. indologenes, E. anopheles*[48] and, *Avrilella dinanensis* that was first isolated from the blood of a septic patient in 2016[49]. *C. indologenes*, as an environmental bacterium, can potentially survive in harsh environments as a unique heat-stable metalloprotease with a non-typical substrate recognition profile that was purified from *C. indologenes* previously[50]. *C. indologenes* has become an emerging opportunistic pathogen that is associated with various types of infections, including sepsis and pneumonia, amongst others[6]. Similar to *C. indologenes, E. anopheles* is also ubiquitously distributed in natural environments, including soil, water, and plants[51] and has become an emerging hospital-acquired pathogen which caused multiple outbreaks around the world due to it being intrinsically multidrug resistant[48,52–54].

Among the homologues of CIM-1, more than half of the top 100 homologues were predicted to be lipoproteins (Lipoprotein Likelihood > 50%). This indicates that lipidated MBLs are potentially much more common than previously assumed. We also observed a lower amino acid similarity in the signal peptides as compared to the mature protein sequences between the homologues. A previous genome-wide analysis study of secretory proteins found that signal peptides evolved faster than mature protein sequences[55]. According to this study, the contributing factor behind the higher evolutionary rate of signal peptides is relaxed selection, where instead of a specific amino acid identity, the general property of signal peptide is preserved. In our study, we observed two pairs of CIM-1 homologues in *Pedobacter sp.* and *E. anopheles,* respectively, where the protein homologue with more speciation events has become lipidated. It is not clear whether higher evolutionary rates of signal peptides are thought to generate more lipidated MBLs nor whether lipidated MBL will provide bacteria with any advantage. More studies are needed to investigate the abundance of this special group of proteins.

Though CIM-1 is a chromosomally encoded MBL and to our knowledge not found on a mobile genetic element yet, many potent resistance mechanisms have originated as chromosomally encoded genes before mobilisation and dissemination. Even *bla*$_{NDM}$ is thought

**Fig. 4 | The non-canonical lipobox is commonly observed in environmental bacteria.** Phylogenetic tree of top 100 BlastP search results of CIM-1 homologues. Lipoprotein likelihood was predicted by SignalP6.0[27]. Lipobox sequences were stated on the right for proteins with a likelihood of being a lipoprotein equal or higher than 50%. Red indicates the same lipobox sequence with CIM-1 (FLNC). Orange indicates three out of four amino acid identities with CIM-1. Yellow indicates two out of four amino acid identities. Black indicates the only conserved amino acid is Cys. WP_123865086.1 has 100% sequence coverage and 100% identity compared to CIM-1. Bacteria species with *: No signal peptide was detected.

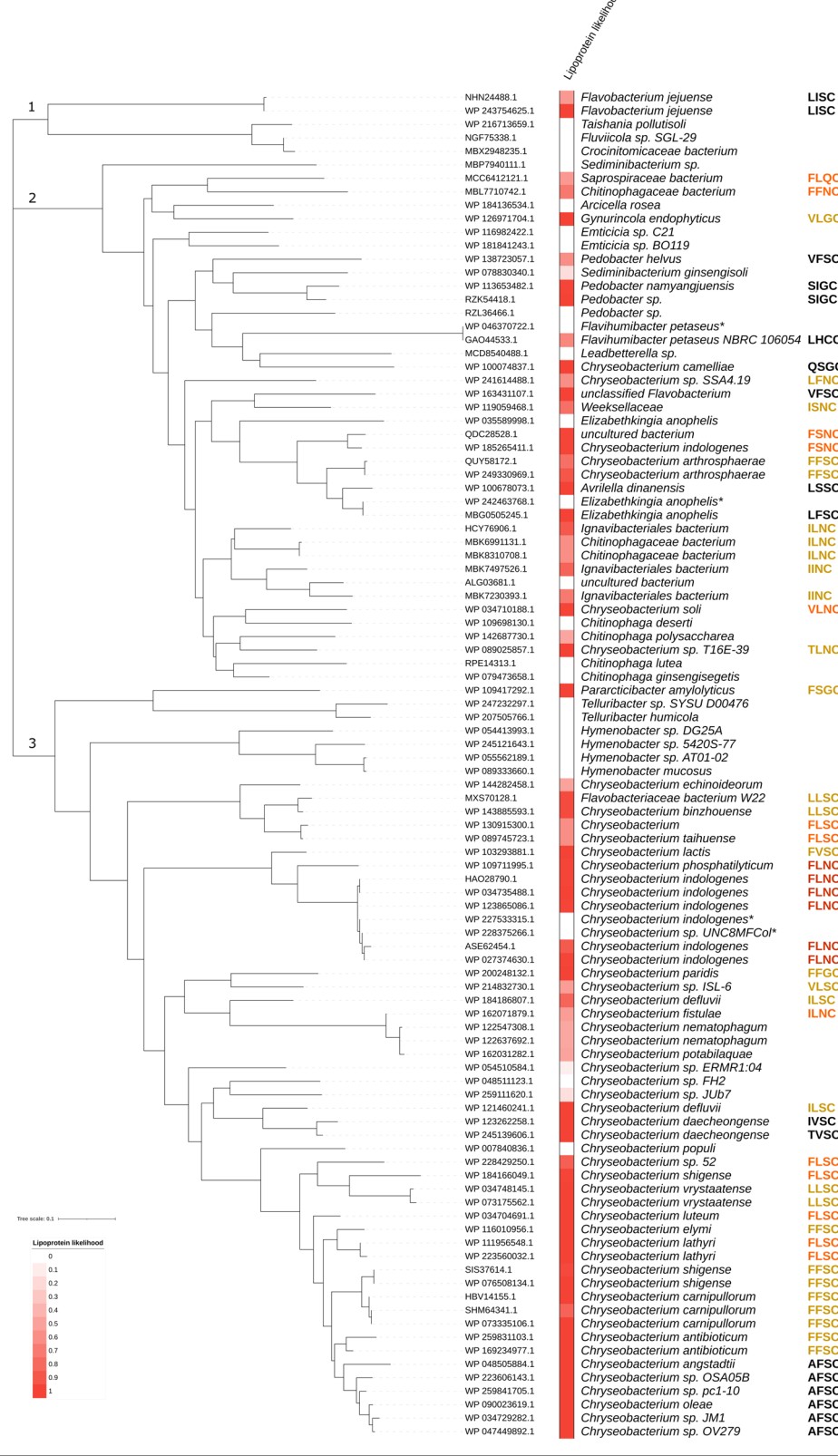

to originate from *Acinetobacter* through a fusion of an ancestral carbapenemase and aminoglycoside resistance gene[56]. From here, it was mobilised by transposition to evolve in the widely disseminated genes that are observed in many different species today[57]. Another example of this phenomenon is another MBL, the Adelaide imipenemase (AIM-1), that originated from a non-pathogenic environmental bacterium, *Pseudoxanthomonas mexicana*[58], before being mobilised to the ubiquitous hospital-acquired pathogen *P. aeruginosa*. The native host of CIM-1, *C. indologenes*, is an environmental organism and opportunistic pathogen found in healthcare environments where this organism's intrinsic resistance to last-resort antibiotics such as carbapenems and colistin could select for *C. indologenes*. The potential for CIM-1 to be mobilised to pathogenic bacteria sharing that environment is, therefore, not improbable.

In conclusion, we have identified and characterised a subclass B1 MBL, with a high affinity for all carbapenem antibiotics from an emerging opportunistic pathogen. This is potentially the first lipidated MBL with a non-canonical lipobox to be identified and characterised. We have also shown that that lipidated MBLs might be much more numerous than originally thought, highlighting the necessity in further investigation of lipidated MBL.

## Methods

### Bacterial strains and reagents

*Chryseobacterium indologenes* #3362 was previously isolated from environmental samples obtained from a residential aged care facility as part of a project on the surveillance of the development and dissemination of AMR in residential aged care facilities. *Klebsiella pneumoniae* D53 is a clinical isolate confirmed to carry NDM-4 and was a kind gift from Ms Jan Bell from AGAR (Australian Group for Antimicrobial Resistance). *Escherichia coli* XL10-Gold (Agilent) was used for the creation of constructs and plasmid propagation. *E. coli* BL21(DE3) was used for protein expression and purification. *E. coli* C41(DE3) cells[59] were transformed with pET41a(+) carrying different constructs and used for protein localisation assays and minimum inhibitory concentration (MIC) determinations. Unless otherwise stated, all strains were grown aerobically at 37 °C in Difco™ Luria-Bertani broth, Miller (BD) supplemented with kanamycin 25 μg/mL. Antibiotics were purchased from Glentham Life Sciences and Sigma-Aldrich. Chemical reagents and oligonucleotides were purchased from Sigma-Aldrich. Enzymes used in cloning were purchased from New England Biolabs.

### Cloning and construction of plasmids and mutants

The plasmid pET41a(+) (Novagen) was used for the creation of all constructs. Primers (Table 4) were used to amplify full-length $bla_{IND-2}$, $bla_{CIM-1}$, and $bla_{NDM-4}$ from *C. indologenes* #3362 and *K. pneumoniae* D53. NDM-4 carries a M154L mutation. PCR products were digested by *Nde*I and *Xho*I (New England Biolabs) and ligated into the pET41(+) vector generating pCIM-1, pIND-2, and pNDM-4. To allow consistent soluble protein expression and purification, signal peptide deletion constructs, CIM-1ΔSig, IND-2ΔSig, and NDM-4ΔSig were also generated. A CIM-1 C19A variant and signal peptide exchange constructs were also generated. All constructs were confirmed by Sanger Sequencing (Australian Genome Research Facility).

### Antimicrobial susceptibility assay

The in vitro minimum inhibitory concentration (MIC) of β-lactams including penicillins, cephalosporins, and carbapenems in *C. indologenes* 3362 and *E. coli* C41(DE3) carrying different constructs were determined using the broth microdilution assay according to the International Standard ISO 20776-1 as recommended by EUCAST (the European Committee on Antimicrobial Susceptibility Testing). Briefly, bacterial cultures were grown aerobically at 37 °C to an $OD_{600}$ 0.4–0.6 in the presence of 25 μg/mL kanamycin, when necessary, before 1 mM IPTG (isopropyl-β-d-thiogalactopyranoside) was added. The induction was allowed for 2 h at 37 °C before the bacterial culture was diluted to $4 \times 10^6$ CFU/mL, and the microbroth dilution assays were performed. Cation-adjusted Mueller Hinton (CAMH) broth was used. All assays were repeated at least four times on four different days with different batches of cells.

### His-tagged protein purification and detection

*E. coli* BL21 cells expressing the signal peptide deletion constructs were grown in 800 mL of LB broth supplemented with kanamycin (50 μg/mL) at 37 °C until they reached an $OD_{600}$ of 0.4–0.6. Following induction with 1 mM IPTG, cultures were incubated for 20 hours at 25 °C. Cells were harvested by centrifugation at $5000 \times g$ for 30 min at 4 °C and resuspended in desalt buffer (50 mM Tris, 200 mM NaCl, 10% w/v Glycerol, pH 7.2) supplemented with 1 μg/mL DNase I (Sigma-Aldrich) and 1× cOmplete™, EDTA-free Protease Inhibitor Cocktail tablet (Roche). Cells were then lysed

by the Constant System cell disrupter (Constant System Ltd) at 35 kpsi and insoluble material was removed by Optima XPN Ultracentrifuge (Beckman Coulter, Australia) for 50 min at $200,000 \times g$ and 4 °C. The 8× His-tagged proteins were purified by immobilised metal affinity chromatography (IMAC) using ÄKTA (Cytiva) protein purification system. The supernatant obtained from ultracentrifugation was loaded onto 1 mL HisTrap™ (Cytiva) column. Unbound and loosely bound proteins were removed was washing with 5 column volumes of buffer (50 mM Tris, 200 mM NaCl, 10 mM imidazole, 10% w/v Glycerol, pH 7.6) before the protein was eluted by a stepwise imidazole gradient of 200, 300, 500 mM in the same buffer. Fractions containing purified protein were pooled, and the buffer was exchanged into desalt buffer using HiPrep™ 26/10 desalting column (Cytiva) to remove the imidazole. All fractions were assessed with SDS-PAGE on a 15% gel followed by Coomassie staining with a purity of ~98%. The concentration of the purified protein was determined using the DC protein assay (Bio Rad).

### Cellular fractionation and protein detection

*E. coli* C41(DE3) expressing CIM-1, IND-2, NDM-4, CIM-1ΔSig, CIM-1 C19A, I-CIM, N-CIM, C-NDM and C-IND proteins were inoculated from an overnight culture and grown in 50 mL LB broth at 37 °C until the cells reached $OD_{600}$ 0.4–0.6. IPTG (0.1 mM) was then added, followed by 2 h incubation at 25 °C. Intact cells were obtained by centrifugation at $7000 \times g$ and resuspended in 50 mM Tris, 200 mM NaCl, 10% w/v Glycerol, pH 7.2. Cells were lysed by the Cell disrupter (Constant system) at 35 kpsi for two passages. Cell debris was removed by centrifugation at $14,000 \times g$ and 4 °C for 20 min. The membrane fraction was obtained from resuspension of the pellet after ultracentrifugation at $200,000 \times g$ and 4 °C for 1 h.

Protein levels were determined by SDS-PAGE followed by Western blot with THE™ His tag monoclonal mouse antibodies (at 1: 10,000 dilution from 0.5 mg/ml solution) and Invitrogen™ Goat anti-Mouse Immunoglobulin G (at 1:5000 dilution). Novex™ ECL Chemiluminescence Substrate Reagent Kit was used according to the manufacturer's instruction to allow for the detection of His-tagged protein on the PVDF membrane. The protein concentration of each fraction was measured using Bio-Rad DC assay and normalised for immunoblotting. AquaStain (Bulldog-Bio) was used to assess protein normalisation.

### Isolation of outer membrane vesicles

*C. indologenes* #3362 culture was grown aerobically at 37 °C in LB broth supplemented with 16 μg/mL meropenem overnight with agitation. Intact cells were removed through centrifugation at $7000 \times g$ for 30 min. The supernatant was filtered through a 0.2 μM cellulose nitrate filter (Sartorius). The filtrate was ultracentrifuged at 150,000 x g for 3 hours to harvest the OMVs, which are then dissolved in phosphate buffered saline (PBS). Resuspended OMVs were plated on LB agar to confirm the complete removal of intact cells. Isolated OMV samples were inspected under TEM.

### Transmission electron microscopy

A volume of *C. indologenes* with an $OD_{600}$ of 1.0 was centrifuged at 13,000 rpm for 1 min (MiniSpin®, Eppendorf). The resulting cell pellets were washed thrice in 1 mL PBS for 1 min. After the final wash and centrifuge as before, the cell pellets were resuspended in 100 μL of fixative (2.5% glutaraldehyde, and 4% paraformaldehyde in PBS, pH 7.2). The fixation was allowed for 1 hour at room temperature. The cell pellet was centrifuged as before and washed thrice in 1 mL PBS. The fixed cells were resuspended in 100 μL PBS at 4 °C until needed. Fixed bacterial cell samples (5 μL) or OMV samples (5 μL) were collected onto carbon-coated 200 mesh copper grids and stained with 2% (w/v) uranyl acetate. Samples were observed under a TEM at 100 kV.

### Steady-state enzyme kinetics

The enzymatic activities of 100 nM NDM-4ΔSig, CIM-1ΔSig and IND-2ΔSig, and towards various β-lactam antibiotics were determined at room temperature ($25 \pm 2$ °C) in 50 mM HEPES (pH 7.0) supplemented with

**Table 4 | PCR primers used in this study with restrictions sites underlined**

| Primer | Oligonucleotide Sequence (5' to 3') |
|---|---|
| *Amplification of bla$_{IND-2}$, bla$_{CIM-1}$ from C. indologenes #3362 DNA extraction* | |
| CIM-1 NdeI$_{Fw}$ | GTTGTTCATATGAAGTCCGTATCCCAAATAC |
| CIM-1 XhoI$_{Rv}$ | GTTGTTCTCGAGTGATTCCATCCTTGAAAAGGGC |
| IND-2 NdeI$_{Fw}$ | GTCGTCCATATGAAAAAAGTATTCAGCTTTG |
| IND-2 XhoI$_{Rv}$ | GACGACCTCGAGTTCCGGCTTTTTATTCTTATC |
| *Amplification of bla$_{NDM-4}$ from K. pneumoniae DNA extraction* | |
| NDM-4 NdeI$_{Fw}$ | ATAAATCATATGGAATTGCCCAATATTATGCACCC |
| NDM-4 XhoI$_{Rv}$ | ATAAATCTCGAGGCGCAGCTTGTCGGCC |
| *Removal of signal peptide from pET41a(+)-bla$_{CIM-1/IND-2}$* | |
| CIM-1ΔSigNdeI$_{Fw}$ | GTTGTTCATATGTGCAATACCAAGAAACCTTCAC |
| CIM-1ΔSigNdeI$_{Rv}$ | GTTGTTCATATGCATATGTATATCTCCTTCTTAAAGTTAAAC |
| IND-2ΔSigBamHI$_{Fw}$ | GTTGTTGGATCCATGCAGGTTAAAGATTTTGTAATTGAG |
| NDM-4ΔSigNdeI$_{Fw}$ | ATAAATCATATGTGCATGCCCGGTGAAATCC |
| *Mutagenesis of CIM-1 Cys19 to Alanine from pET41a(+)-bla$_{CIM-1}$* | |
| CIM-1 C19A$_{Fw}$ | CTGTTTTTATTTTTTTGAATG**GCG**AATACCAAGAAACCTTC |
| CIM-1 C19A$_{Rv}$ | GAAGGTTTCTTGGTATT**CGC**ATTCAAAAAAATAAAAACAG |
| *Signal peptide exchange constructs using pET41a(+)-bla$_{CIM-1}$, pET41a(+)-bla$_{IND-2}$, and pET41a(+)-bla$_{NDM-4}$* | |
| I-CIM BamHI$_{Fw}$ | GTGTTGGATCCATGAAAAAAGTATTCAGCTTTGATGATGTCAATGTTTTTAAGCCCATTGATCAATGCCCAGGTTAAAAAGAAACCTTCACATGTGCC |
| C-IND BamHI$_{Fw}$ | TTGTTGGATCCATGAAGTCCGTATCCCAAATACTATTACTTTCCCTGTTTTTATTTTTTTGAATTGCAATACCGATTTTGTAATTGAGCCGCCTG |
| N-CIM BamHI$_{Fw}$ | GTGTTGGATCCATGAATTGCCCAATATTATGCACCGGTCGCGGAAGCTGAGCACCGCATTAGCCGCTGCATTGATGCTGAGCGGCGGGTGCATGCCC AAGAAACCTTCACATGTGCC |
| pETBamHI$_{Rv}$ | GTGTGGGATCCATCTCCTTCTTAAAGTTAAACAAAATTATTTC |
| C-NDM NdeI$_{Fw}$ | ATAAATCATATGAAGTCCGTATCCCAAATACTATTACTTTCCCTGTTTTTATTTTTTTGAATGCAATACCGGTGAAATCCGCGCCGACG |

10 μM ZnSO$_4$ using a Cytation5® (Bio-Tek®) plate reader. Analyses were carried out at least in duplicate at a final volume of 100 μL. A protein-only reaction was used as a negative control. As a control for inhibition of β-lactamase activity, 100 μM of the Zn$^{2+}$ chelator, 1,10-phenanthroline was added 20 minutes prior to the measurements. For nitrocefin-dependent assays, absorbance at 486 nm was measured as an indication of hydrolysed nitrocefin formation and absorption coefficients (20,500 M$^{-1}$ cm$^{-1}$) in a flat-bottom 96-microwell plate (Costar®). For the remaining antibiotic agents, a standard curve (concentration vs. absorbance) was plotted for each antibiotic and was used to calculate the substrate concentration. Imipenem and meropenem were detected at 300 nm or 320 nm and ceftazidime and cefepime at 260 nm using UV transparent 96-well plates (Greiner UV-Star®). *Km* and *kcat* values were determined under initial rate conditions using non-linear regression curve analysis performed using GraphPad Prism v. 9.2.

## In-gel digestion for mass spectrometry analysis

Isolated OMVs and membrane fractions of *C. indologenes* were run on a 15% SDS-PAGE gel for 10 min before immersion in AquaStain (Bulldog Bio). Excised gel bands were crushed and de-stained twice in 30% ethanol for 30 min. After supernatant removal, in-gel digestion was performed according to nature protocol[60], except 2-chloroacetamide was used for protein alkylation instead of iodoacetamide. The extracted tryptic peptides were dried down and desalted with ZipTips (Merck Millipore) before storage at −20 °C. LC–MS analysis on duplicate peptide samples (1 μg) was conducted on an EASY-nLC 1200 (Thermo Scientific) coupled to an Orbitrap Exploris 480 mass spectrometer (Thermo Scientific) by Mass Spectrometry and Proteomics Facility, University of South Australia.

## Reverse-transcription quantitative polymerase chain reaction (RT-qPCR)

The relative level of mRNA expression of CIM-1 and IND-2 when *C. indologenes* was cultured with meropenem (16 μg/mL) was determined using KAPA SYBR FAST One-Step RT-qPCR Kit (Sigma-Aldrich) following the manufacturer's instructions and Mic real-time PCR cycler (Bio Molecular Systems). Primers *bla*$_{CIM-1}$_F 5′-TAGGAGCAGGCAAAGGCTAC-3′ and *bla*$_{CIM-1}$_R 5′-GTATCACCATGACCCGGGATG-3′, *bla*$_{IND-2}$_F 5′-TGCTGTATTTGCCACCCACT-3′ and *bla*$_{IND-2}$_R 5′-TCTTGGCGGTCGCATATGTT-3′, Rho_F 5′-AACGGACGTGATCTTGCCTTC-3′ and Rho_R 5′GATCCGCTTCCTGCAAGA-3′ were used.

## DNA extraction and whole-genome sequencing

Genomic DNA was extracted using the MN NucleoSpin® Microbial DNA kit (Machery-Nagel) following the manufacturer's instructions. Whole-genome sequencing was carried out at SA Pathology (Adelaide, SA, Australia) using the Illumina NextSeq 550 platform with the NextSq 500/550 Mid-Output kit v2.5 (300 cycles) (Illumina). Sequencing libraries were prepared using Nextera XT DNA Library Preparation Kit.

## Bioinformatic analysis

Raw paired-end sequencing reads were assembled and annotated using the TORMES pipeline v.1.2[61]. The draft genome was assembled using SPAdes[62] and annotated by Prokka[63]. Multiple protein sequence alignments were carried out using Clustal Omega[64] and visualised with the ESPript 3.0 server[41]. Signal peptides were assessed using SignalP-6.0[27]. SWISS-MODEL Server[33] was used to generate 3D model of CIM-1. Visualisation was carried out with the use of Chimera[65]. Basic local alignment search tool (BLAST) analysis was performed using the amino acid sequence of CIM-1 (accession number: WP_123865086.1) as a query search. A phylogenetic tree was generated using the maximum likelihood method tested with the bootstrap method (1000 replicates) with MEGA (Molecular evolutionary genetic analysis)[66] and visualised by iTOL[67].

## Statistics and reproducibility

Enzyme kinetic data was obtained from three individual experiments. Michaelis–Menten kinetic analysis was performed using GraphPad Prism

9.2. Data represented as mean ± standard error of the mean (SEM). Cellular fractionation and Western blot were performed in $n = 3$ biologically independent samples.

## Reporting summary

Further information on research design is available in the Nature Portfolio Reporting Summary linked to this article.

## Data availability

Whole genome sequence of *C. indologenes* #3362 was deposited into the GenBank, National Centre for Biotechnology Information, under accession number JAPSGE010000000. Source data underlying the kinetic results (Table 3) can be found in Supplementary Data 1. Uncropped blots can be found in Supplementary Fig. 11. Source data used to generate Table 2 can be found in Supplementary Data 2. Source data used to generate the phylogenetic tree (Fig. 4) can be found in Supplementary Data 3. All other data are available from the corresponding author upon reasonable request.

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

## Acknowledgements

The authors would like to thank Ms Jan Bell from the Australian Group for Antimicrobial Resistance for the gift of the *K. pneumonia strain* producing NDM-4. The authors would also like to extend their gratitude to the aged care provider for allowing access to the residential aged care facility where the swabs were obtained and to Mr Barry Lowe for performing sampling in the aged care facility. The authors acknowledge Bioplatforms Australia and the State and Federal Governments, which co-fund the NCRIS-enabled Mass Spectrometry and Proteomics Facility at the University of South Australia. The authors also would like to thank Dr Clifford Young for performing pro-teomic analysis and Christopher Leigh from Adelaide microscopy for his assistance with the Transmission electron microscopy. This work was fun-ded by the Medical Research Future Fund (MRFF, GN1152556) to HV. YW is the recipient of a University President's Scholarship (UPS) from the Uni-versity of South Australia.

## Author contributions

Y.W., S.A.S, and H.V. conceived the project and designed the experiments. Y.W. wrote the original draft of the paper and prepared the figures. Y.W, S.A.S., and J.J.W performed the experimental work. J.M.B. isolated *C. indologenes* #3362 from the environmental sample and performed the preliminary antimicrobial susceptibility test. S.A.S. and O.L., edited the paper. H.V. supervised the project, provided the funding acquisition, and conceptualised and edited the paper. All authors reviewed the paper.

## Competing interests

The authors declare no competing interests.
