## [Peer Review File · Communications Biology]

Reviewers' comments:

Reviewer #1 (Remarks to the Author):

Wang et al describe the identification of a new metallo-beta-lactamase (MBL) harboured by the opportunistic pathogen *Chryseobacterium indologenes* (named CIM-1). The authors highlight that this is only the second described lipidated MBL after the well-described NDM-1 enzyme. Amino acid sequence analysis and modeling revealed low similarity with NDM-1; however, CIM-1 has highly conserved residues in the zinc binding site of class B1 MBLs. The authors report that when CIM-1 was expressed in *E. coli*, the enzyme was able to confer resistance to all the antibiotics tested. They also performed cellular localization studies revealing the enzyme is membrane associated, including mutagenesis of an amino acid (C19) proposed to be important for lipidation. Additionally, the N-terminal region of CIM-2 and other MBLs were altered to see how localization was impacted. Enzyme assays revealed CIM-1 exhibits broad spectrum affinity for the beta-lactam substrates. The study was concluded with a bioinformatic analysis assessing the incidence of 'lipidation' in MBL CIM-1 homologs. Overall, this appears to be an interesting new MBL, and the work is important given the association of MBLs with beta-lactam resistance in the clinic. To support the major conclusions of the study, below are comments for consideration.

An overarching conclusion of this study is that CIM-1 and homologs are lipidated. However, the authors reveal that CIM-1 has a non-canonical lipobox and mutagenesis of the only conserved residue, the cysteine, did not abolish membrane localization (Fig. 2) and did not affect susceptibility (Table 1). The conserved lipobox sequence, [LVI][ASTVI][GAS]C, is for the most part absent from CIM-1. The authors suggest that other residues could be associated with electrostatic membrane interactions, although it is not clear which residues they are referring to (Line 329). This aspect of the work could be strengthened to support the conclusions of the study. The included immunoblots are very closely cropped (it is advisable to include at least two molecular weight markers) and the entire gel should be included in the supplementary information. It is not clear if loading controls were used and the nature of the replicates should be described, in addition to protein abundance quantification. Since these were only n=2 (independent measurements?), the individual data points should be shown.

The authors report that when CIM-1 was expressed in *E. coli*, the enzyme was able to confer resistance to all the antibiotics tested (line 192, Table 1). The data do not support this statement, in some cases the increase in resistance is only 2-fold, which is within the acceptable error range for susceptibility testing. The nature of the replicates should be described, and fold changes shown. The materials and methods for susceptibility testing should be more detailed.

Extensive mutagenesis of the N-terminal CIM-1 region was undertaken to assess localization and activity. I found the construct nomenclature to be a little confusing and Figure 3d is not easy to interpret; what does the 'expression level' refer to seeing as whole cell lysate is assessed? Surprisingly, fusing the signal peptide of NDM-1 to CIM-1 enhanced resistance considerably. Fusion of the CIM-1 signal peptide to NDM-1 reduced resistance. Finally, removal of the CIM-1 signal peptide conferred high-level resistance;

the localization of this construct was not assessed (Fig. 3d) and would have functioned as a useful control. The comments made above regarding the immunoblots are appropriate here too. Overall, protein abundance appears to be impacting the susceptibility testing and the N-terminal region is affecting abundance. The results do seem to support membrane localization being associated with the N-terminal regions of CIM-1 and NDM-1, especially since C-IND was present in the membrane fraction. However, this reviewer was wondering whether these proteins are membrane localized or simply insoluble/aggregated/unfolded when altered in this way. It would be useful if the authors could comment on this possibility. Localization studies of CIM-1 should perhaps be undertaken in the native host (*Chryseobacterium indologenes*). Did the authors attempt to disrupt the CIM-1 gene and/or the other MBL-encoding genes and see if susceptibility of the *C. indologenes* isolate was affected?

The figures could be improved since many of the structures are very small and difficult to interpret. The introduction would also benefit from more description of *Chryseobacterium* spp for those who are unfamiliar. It is suggested the authors expand on the lipobox and include greater detail about studies describing the lipobox of NDM-1.

Minor comments:

Line 14 – broad statement. I would suggest adding ‘clinically utilized.’

Line 73 – worth mentioning lipidation also enables NDM-1 to be secreted in outer membrane vesicles

Line 113 – how does this compare to an AlphaFold model?

Line 152 – this does not conclusively show the protein is membrane-associated through lipidation. Did the authors attempt to purify from the producing organism?

Line 198 – better explain N-CIM and C-NDM

Line 219 – unknown if this is an electrostatic interaction

Line 323 - cysteine

Line 327 – moiety

Line 345 – genes are expressed

Line 462 – insufficient information for susceptibility testing. Cell inocula can have significant effects on beta-lactam susceptibility and the dilution used (CFUs in final) should be documented. Mueller Hinton cation adjusted broth is also recommended for susceptibility testing and duration of incubation should be included.

Line 481 – desalt buffer?

Line 483 – the purity of the protein preparations should be described since this is important for the kinetic enzyme assays.

Line 503 – protein concentration should be listed

Reviewer #2 (Remarks to the Author):

This paper describes the identification and characterization of CIM-1, a novel membrane-bound carbapenemase involved in multidrug resistance in the opportunistic bacterium *Chryseobacterium indologenes*. This is somewhat of a novelty. It is probably a good job in that they have done a lot of experiments, including making various mutants and measuring enzyme chemistry, and have examined CIM-1 closely.

However, the finishing touches in the paper are so poor and crude. The data is inconsistent in some places, and I almost doubt that the content is even okay.

I think that since the authors have done a lot of work, the paper would be much better if the authors paid more proper attention to detail.

I will list the points that I found problematic, but first of all, I have the impression that the results are very poorly presented in this paper. It could have been a very good paper with novel content, but the paper is so poorly finished that I might happen to doubt the content.

First of all, this may not be something to blame the authors for, but the text on L.217 is incomplete. It may be a system problem, but the beginning of line 217 does not connect to anywhere before it. However, this is not enough to obscure the entire paper, nor is it critical to change the impression.

L.50

Roman B should be Greek beta.

L.128

Table Fig.1a does not match the MIC values in Table.1. Perhaps the MIC measurements performed in this study are those listed in Table.1, but what is the origin of the table in Fig.1a? There is no citation, and I cannot understand. Is it necessary?

L.172

The scale of the vertical axis is significantly different between Fig.2b and c. Is this a problem in the interpretation of the data in this study?

L.211

Fig.3d, there are lanes for Expression Level and Whole Cell lysate. What is the difference between them? I can't figure it out from reading the Materials and Methods.

In some lanes, there is almost no expression (CIM-1 is a notable example), however the whole cell lysate and other fractions have protein bands, which is difficult to understand. Is it necessary to show both? How about reexamining the results, including the question of whether it is necessary to show both?

L.211

Furthermore, about this Fig. 3d,

A panel for CIM-1Δsig is not shown, which I think is necessary. Also, there is a description in the

Materials and Methods, so perhaps the authors simply forgot to include it?

L.227

The referring 3c on L.227, but my impression from what I have read is that this should be 3d?

L.231 and another lines

There should be uniformity in the way proteins and mutants are described.

L.231, it is CIM, but mostly CIM-1 are used. It would be more reader-friendly to unify the writing style.

There is also another inconsistency on L.486 as well. In Supplementary Fig. 5, there are many. CIND / C-IND, ICIM / I-CIM. They should be fixed.

Efforts should be made to unify this as it makes it difficult to understand. Even scientific content might be underestimated.

L.233

Also, 3b should be 3d?

L.306.

I think it's fair to say that CIM-1 is identified, but I don't feel that IND-2 is identified in this paper. Isn't it enough that this paper has one significant study of CIM-1, as the title says? Or is there not much more to IND-2 than the Nordmann et al. paper in 2000? If there is an argument you want to emphasize, you should add a little more to that section.

L.353

The OMV appears for the first time in this paper as an abbreviation, so it should not be abbreviated here.

L457

The authors described in the Materials and Methods that BC21(DE3) is used in this study. Where is this cell used? Looking at Table.2, it seems that it is all expressed in C41(DE3).

L.482

There is a difference between the mutants listed in L486 and those listed in Fig3. Wouldn't they be the same?

Supplementary data L.45

Isn't CIM-1 in Supplemental Fig. 3 also CIM-1 Δ Sig?

Supplementary data L.45

Supplemental Fig. 3 shows the actual curves for the measurements summarized in Table-2, but why is only this one selected to be shown? Couldn't you also show data for other drugs or other series of mutants? It may be too much to show one by one, but it could be organized well by overlaying the three

proteins (including mutants) for each drug with different markings (▲, etc.).

Supplementary data L.67

L.67 of the supplementary information, they have the descriptions as CIM-1, CIM-1 C19A just above the two gel pictures, but the way it is written from line 68, shouldn't it be written on the bottom edge of the gel? And instead, there are no explanations for each lane, so I cannot understand what is being loaded.

Supplementary data L.69

Why is there a solid band on the Empty lane?

It appears more intense than the band of Expression levels of CIM-1 in Fig. 3d.

Is this paper okay? I'm just wondering.

I think that since you have done a lot of work, the paper would be much better if you paid more proper attention to detail.

Also, CIM-1 C19A on the right side of line 67 is the same figure as the left half of CIM-1 C19A on line 68, second from the left (in that sense, it seems like a useless duplication). However, the CIM-1 on the left side of line 67 is different from the CIM-1 on the left side of line 68. Why are these twice the same but different? I have a hard time understanding this. I may be going to be suspicious of this paper itself. Is it a simple mistake?

My conclusion is that I believe that this work has sufficient potential to deserve recognition, but the paper is apparently poor. Please, to value the effort, time and resources spent, take care of the presentation of the data.

Reviewer #3 (Remarks to the Author):

The work of Wang et al. presents the identification and characterization of a lipidated metallo- β -lactamases, CIM-1, from the opportunistic pathogen *Chryseobacterium indologenes*. CIM-1 is the second reported lipidated MBL and possesses an atypical lipobox that could potentially influence its expression, localization and activity. Overall, this study provides valuable insights into the characterization of CIM-1. However, to enhance the impact and broaden the interest of this work, it could benefit from additional functional characterization of the atypical lipobox and further evidence supporting the potential evolutionary advantages associated with this unique sequence.

Major points:

1. The authors demonstrated the hydrophobic interaction between CIM-1 and the membrane by showing its solubilization with Triton X-100. However, it remains unclear whether CIM-1 associates with the inner membrane or the inner leaflet of the outer membrane. Considering the authors' speculation regarding CIM-1 secretion in OMVs, it is crucial to confirm whether CIM-1 is associated with the outer membrane. Additionally, does CIM-1 possess any signal to avoid the LolCDE translocation system?

2. There appears to be missing content before lane 217. Moreover, the “expression levels” presented in Fig 3.d are somewhat perplexing. Are these western blots performed using antibodies against any specific tags? If not, comparing the expression levels of different lactamases could be challenging due to potential differences in antibody efficiency. Furthermore, the “soluble fraction” of N-CIM exhibits a smaller molecular weight than the band in the membrane fraction. Also, two bands can be seen in the soluble fraction of I-CIM. Could these bands be a result of proteolysis of the membrane-bound CIM?
3. The authors proposed that CIM-1 C19A associates with membrane through electrostatic interactions. It would be valuable to investigate the surface properties of CIM-1 based on the structural model. Does CIM-1 possess a surface area with positive electrostatic potential that might facilitate its membrane association?
4. It is evident that the presence of the atypical lipobox sequence significantly affects the level of CIM-1 when recombinantly expressed in E.coli. It remains unclear whether the same effect is observed in C. indologenes. It would be intriguing to explore whether this atypical yet conserved lipobox sequence confers any evolutionary benefits. The authors raised the intriguing possibility that CIM-1 could be excreted into the media through OMVs. Is it feasible to experimentally test this hypothesis and compare the secretion efficiency of CIM-1 with typical or atypical lipobox sequences? Studies have indicated that membrane anchoring contributes to the stability of NDM-1 (Nat Chem Biol. 2016 Jul;12(7):516-22), could the atypical lipobox sequence play a similar role in CIM-1 stability?
5. It is essential to indicate the number of replicates performed for the MIC and western blot experiments, as well as specify the antibodies used.

Reviewers' comments:

Reviewer #1 (Remarks to the Author):

Wang et al describe the identification of a new metallo-beta-lactamase (MBL) harboured by the opportunistic pathogen *Chryseobacterium indologenes* (named CIM-1). The authors highlight that this is only the second described lipidated MBL after the well-described NDM-1 enzyme. Amino acid sequence analysis and modeling revealed low similarity with NDM-1; however, CIM-1 has highly conserved residues in the zinc binding site of class B1 MBLs. The authors report that when CIM-1 was expressed in *E. coli*, the enzyme was able to confer resistance to all the antibiotics tested. They also performed cellular localization studies revealing the enzyme is membrane associated, including mutagenesis of an amino acid (C19) proposed to be important for lipidation. Additionally, the N-terminal region of CIM-2 and other MBLs were altered to see how localization was impacted. Enzyme assays revealed CIM-1 exhibits broad spectrum affinity for the beta-lactam substrates. The study was concluded with a bioinformatic analysis assessing the incidence of 'lipidation' in MBL CIM-1 homologs. Overall, this appears to be an interesting new MBL, and the work is important given the association of MBLs with beta-lactam resistance in the clinic. To support the major conclusions of the study, below are comments for consideration.

1. An overarching conclusion of this study is that CIM-1 and homologs are lipidated. However, the authors reveal that CIM-1 has a non-canonical lipobox and mutagenesis of the only conserved residue, the cysteine, did not abolish membrane localization (Fig. 2) and did not affect susceptibility (Table 1). The conserved lipobox sequence, [LVI][ASTVI][GAS]C, is for the most part absent from CIM-1. The authors suggest that other residues could be associated with electrostatic membrane interactions, although it is not clear which residues they are referring to (Line 329). This aspect of the work could be strengthened to support the conclusions of the study.

Thank you for your comment. We agree with the reviewer that mutating the conserved Cys residue does not abolish membrane interaction. This interaction could partly be due to electrostatic interactions offered by the matured protein sequence as was shown by Alessio Prunotto *et al.* for NDM-1 (Prunotto, Bahr, Gonzalez, Vila, & Dal Peraro, 2020). However, CIM-1 is expressed in a non-native host harbouring a different Lol system. We can thus not rule out that both wild-type and the CIM-1 C19A proteins might not be properly localised. Therefore, we have followed the suggestion of this reviewer and reviewer 3 and have investigated the membrane localisation in the native host *C. indologenes*. We have performed liquid chromatography-tandem mass spectrometry (LC-MS/MS) on CIM-1 and showed that CIM-1 predominantly localises to the membrane fraction compared to the non-lipidated MBL from *C. indologenes*, IND-2. A new section, Section 4 was added as well as Figure 4 that are also included below for the reviewer's reference.

Figure 4. CIM-1 is identified in the membrane fraction of *C. indologenes*.

a) Relative protein abundance of CIM-1 and IND-2 presented in the membrane fraction of *C. indologenes* #3362 normalised by protein expression level. b) Ratio of protein presented in membrane and outer membrane vesicles isolated from *C. indologenes* #3362

2. The included immunoblots are very closely cropped (it is advisable to include at least two molecular weight markers) and the entire gel should be included in the supplementary information.

The membrane fractionation was repeated several times. Protein loading was normalised by protein quantification using the Bio-Rad DC assay and equal protein loading was indicated by Coomassie staining. A representative entire Coomassie stained gel was included in as Supplementary Figure 12 and is also copied below.

Supplementary Figure 12. Full, uncut gel images for Figure 3b

3. It is not clear if loading controls were used and the nature of the replicates should be described, in addition to protein abundance quantification. Since these were only n=2 (independent measurements?), the individual data points should be shown.

Please see our answer to question 2. In short, equal loading was ensured through protein quantification. An entire Coomassie stained gel is shown in Supplementary Fig 12. The relative expression and membrane fractionation was repeated at least 3 times on different days with different batches of cells. Representative gels are shown for Figure 3b and Supplementary Figure 12.

4. The authors report that when CIM-1 was expressed in *E. coli*, the enzyme was able to confer resistance to all the antibiotics tested (line 192, Table 1). The data do not support this statement, in some cases the increase in resistance is only 2-fold, which is within the acceptable error range for susceptibility testing. The nature of the replicates should be described, and fold changes shown. The materials and methods for susceptibility testing should be more detailed.

The modest change in MIC for CIM-1 against some of the antibiotics is in direct correlation with the low expression level of CIM-1 as shown in Supplementary Figure 4. When the expression level of CIM-1 is higher, as with CIM-1 Δ Sig, the MIC against all the antibiotics tested (apart from aztreonam) is ≥ 128 $\mu\text{g/mL}$, indicating that CIM-1 is indeed able to confer resistance to these antibiotics when expressed in *E. coli*. Moreover, the susceptibility experiment was repeated at least four different times on different days with different batches of cells and the MIC value increase for CIM-1 was observed every time without any exception.

In response to the reviewer's request, we have added the fold changes to Table 1.

In response to the reviewer's request, we have added more detail to the antimicrobial susceptibility testing. This part of the Materials and Methods now reads as shown below (lines 571-578 on page 27):

Antimicrobial susceptibility assay

The *in vitro* minimum inhibitory concentration (MIC) of β -lactams including penicillins, cephalosporins, and carbapenems in *C. indologenes* 3362 and *E. coli* C41(DE3) carrying different constructs were determined using the broth microdilution assay according to the International Standard ISO 20776-1 as recommended by EUCAST (the European Committee on Antimicrobial Susceptibility Testing). Briefly, bacterial cultures were grown aerobically at 37 °C to an OD₆₀₀ 0.4 – 0.6 in the presence of 25 μ g/mL kanamycin when necessary, before 1 mM IPTG (isopropyl- β -d-thiogalactopyranoside) was added. The induction was allowed for 2 hours at 37 °C before the bacterial culture was diluted to 4 x 10⁶ CFU/mL and the microbroth dilution assays were performed. Cation-adjusted Mueller Hinton (CAMH) broth was used. All assays were repeated at least four times on four different days with different batches of cells.

5. Extensive mutagenesis of the N-terminal CIM-1 region was undertaken to assess localization and activity. I found the construct nomenclature to be a little confusing and Figure 3d is not easy to interpret; what does the 'expression level' refer to seeing as whole cell lysate is assessed?

Thank you for your advice. Figure 2 was updated to indicate more clearly what the various constructs are. The expression level of all the different constructs was included as supporting evidence to show that the different MIC values obtained in the susceptibility assays were mainly due to the vastly different expression levels of constructs. To avoid confusion, the blot indicating the expression level was moved to the supplementary material as Supplementary Figure 4.

Expression level of all constructs in *E. coli* C41(DE3)

Supplementary Figure 4. Relative expression level of all His-tagged constructs (CIM-1, IND-2, NDM-4, CIM-1ΔSig, CIM-1 C19A, I-CIM, N-CIM, C-NDM and C-IND) in *E. coli* C41(DE3) cells.

Cells were inoculated from an overnight culture and grown the same manner as preparation for antimicrobial susceptibility assay.

6. Surprisingly, fusing the signal peptide of NDM-1 to CIM-1 enhanced resistance considerably. Fusion of the CIM-1 signal peptide to NDM-1 reduced resistance. Finally, removal of the CIM-1 signal peptide conferred high-level resistance; the localization of this construct include signal detection for DeltaSig-CIM-1 was not assessed (Fig. 3d) and would have functioned as a useful control.

We apologise for this oversight. The localisation of CIM-1 Δ Sig was accessed and included into Figure 3b. The differences in resistance conferred is a direct result of the relative expression level of the different constructs as indicated in Supplementary Figure 4. It is also clear that the presence of the signal peptide from *C. indologenes* hampers expression level of these proteins in *E. coli*. This is perhaps not surprising, given that *C. indologenes* belongs to a different family of organisms than *E. coli* and differences in the protein recognition, folding and trafficking machinery can be expected. The latter is a subject of another study that is currently ongoing in our group.

Figure 3. Schematic representation of constructs and their postulated cellular localisation.

a) Schematic representation of the postulated cellular localization of the constructs. **b)** Representative Western blot images of protein levels of various protein constructs in whole cells and cellular fractions from *E. coli* C41(DE3). THE™ His tag monoclonal mouse antibodies and Invitrogen™ Goat anti-Mouse Immunoglobulin G was used in immunodetection.

7. The comments made above regarding the immunoblots are appropriate here too. Overall, protein abundance appears to be impacting the susceptibility testing and the N-terminal region is affecting abundance. The results do seem to support membrane localization being associated with the N-terminal regions of CIM-1 and NDM-1, especially since C-IND was present in the membrane fraction. However, this reviewer was wondering whether these proteins are membrane localized or simply insoluble/aggregated/unfolded when altered in this way.

It would be useful if the authors could comment on this possibility.

The authors agree with reviewer's comment about the susceptibility being impacted by expression level. We do not think that the proteins are insoluble/aggregated/unfolded though as even at the relatively low expression level of e.g., C-IND a 16x fold increase in the MIC of ampicillin and 4x increase in the MIC values of doripenem and imipenem were routinely observed indicating a correctly folded and active protein. We also expect the low-speed spin (14, 000 xg) that is used after cell breaking to remove insoluble and aggregated proteins as part of the cell debris and inclusion bodies.

9. Did the authors attempt to disrupt the CIM-1 gene and/or the other MBL-encoding genes and see if susceptibility of the *C. indologenes* isolate was affected?

Thank you for your comment. As there is no available literature about genetic manipulation of *C. indologenes*, it is very difficult to perform this experiment in the timeframe as the tools for genetic manipulation of *C. indologenes* must be established first. We also looked for the gene manipulation method for closely relative bacteria such as *Flavobacterium* spp., that revealed that *recB* and *recC* are absent, and RecA dependent homologous recombination is rare in *Flavobacterium* (Staroscik, Hunnicutt, Archibald, & Nelson, 2008). Unfortunately, *recB* and *recC* are also absent in *C. indologenes*. This work falls outside of the remit of this manuscript but is the topic of another study following from the research reported in this manuscript.

10. The figures could be improved since many of the structures are very small and difficult to interpret. The introduction would also benefit from more description of *Chryseobacterium* spp for those who are unfamiliar. It is suggested the authors expand on the lipobox and include greater detail about studies describing the lipobox of NDM-1.

We thank the reviewer for their suggestion to improve the clarity of our manuscript.

Figure 1 has been updated as follows: the structures was enlarged, and the text size used in the figure was increased.

Figure 2: The schematic drawings and all the text in this figure were enlarged for clarity.

We have expanded on the Introduction to include more information regarding *C. indologenes* and the lipobox. The sections shown below were added to the Introduction.

Page 2, lines 40-45:

"Among the six species under this genus, the incidence of C. indologenes infections has been increasing for the past two decades (Alyami et al., 2020; Izaguirre-Anariba &

Sivapalan, 2020). The rod-shaped, yellow-pigmented, Gram-negative bacteria are ubiquitous in environmental soil and water. It can also be found in hospital environments on wet, humid surfaces such as respirators, feeding tubes, and indwelling devices (Mehta & Pathak, 2018)."

Page 3, lines 77-88:

"Additionally, the lipidation of NDM potentially prevent it from degradation under Zn(II) starvation condition and aids secretion of this enzyme in outer membrane vesicles (OMVs) (Gonzalez et al., 2016). Although 43 other MBLs was predicted as lipoproteins in the study performed by Gonzalez et al. (2016) using the lipobox conserved sequence, NDM remains the only characterized MBL to date.

Bacterial lipoproteins can be identified using the consensus sequence of the lipobox located at the C region of the signal peptide. The conserved sequence has expanded from [L][AS][GA][C] based on 26 distinct lipoprotein precursors to [LVI][ASTVI][GAS][C] through the decades as more and more lipoproteins have been identified (Babu et al., 2006; Hayashi & Wu, 1990; Kovacs-Simon et al., 2011). Due to the advances of computer technology, several prediction tools were developed including SignalP 6.0 that uses a machine learning model to detect all known types of signal peptides (Teufel et al., 2022)."

Minor comments:

Line 14 – broad statement. I would suggest adding 'clinically utilized.'

This has been amended, line 13 on page 1

Line 73 – worth mentioning lipidation also enables NDM-1 to be secreted in outer membrane vesicles.

This has been amended, line 79 on page 3

Line 113 – how does this compare to an Alphafold model?

Alphafold was used to predict the structure of CIM-1, the result turns out to be very similar with the previous model built by SWISS-MODEL. SWISS-MODEL now has access to AlphaFold DB as shown in the screenshot below. The 'Compare' function allowed us to observe the differences. The differences were observed around the loops but are quite minor which we believe is due to the vibration of the loop during substrate binding. SWISS-MODEL was used as it is built on experimentally generated models and shows close relatives of CIM-1.

Line 152 – this does not conclusively show the protein is membrane-associated through lipidation. Did the authors attempt to purify from the producing organism?

This has been removed from the manuscript.

Line 198 – better explain N-CIM and C-NDM

We have added more details on the designing of signal peptide exchange constructs from line 221 to 230 on page 9. Figure 2a, c was also amended and enlarged for clarity.

Line 219 – unknown if this is an electrostatic interaction.

This has been removed from the manuscript.

Line 323 – cysteine

This has been amended, line 415 on page 21

Line 327 – moiety

This has been removed from the manuscript.

Line 345 – genes are expressed

This has been amended, line 446 on page 21.

Line 462 – insufficient information for susceptibility testing. Cell inocula can have significant effects on beta-lactam susceptibility and the dilution used (CFUs in final) check but should be something like 1×10^5 CFU/mLs should be documented. Mueller Hinton cation adjusted broth is also recommended for susceptibility testing and duration of incubation should be included.

The materials and methods were updated as indicated below to include all the requested detail. Mueller Hinton cation adjusted broth was used for the susceptibility assays.

lines 571-578 on page 27

“Antimicrobial susceptibility assay

*The in vitro minimum inhibitory concentration (MIC) of β -lactams including penicillins, cephalosporins, and carbapenems in *C. indologenes* 3362 and *E. coli* C41(DE3) carrying different constructs were determined using the broth microdilution assay according to the International Standard ISO 20776-1 as recommended by EUCAST (the European Committee on Antimicrobial Susceptibility Testing). Briefly, bacterial cultures were grown aerobically at 37 °C to an OD_{600} 0.4 – 0.6 in the presence of 25 μ g/mL kanamycin, when necessary, before 1 mM IPTG (isopropyl- β -D-thiogalactopyranoside) was added. The induction was allowed for 2 hours at 37 °C before the bacterial culture was diluted to 4×10^6 CFU/mL and the microbroth dilution assays were performed. Cation-adjusted Mueller Hinton (CAMH) broth was used. All assays were repeated at least four times on four different days with different batches of cells.”*

Line 481 – desalt buffer?

The recipe of desalt buffer is indicated earlier in the paragraph (line 585, page 28) as 50 mM Tris, 200 mM NaCl, 10% w/v Glycerol, pH 7.2.

Line 483 – the purity of the protein preparations should be described since this is important for the kinetic enzyme assays.

The purity of the protein was assessed as ~98% from analysis of the Coomassie stained gel of the purification fractions on SDS-PAGE (Supplementary Figure 5). A note stating the purity of the protein was added to the Results page 29, line 598.

Line 503 – protein concentration should be listed

The protein concentration was added (now in 649, page 29)

Reviewer #2 (Remarks to the Author):

This paper describes the identification and characterization of CIM-1, a novel membrane-bound carbapenemase involved in multidrug resistance in the opportunistic bacterium *Chryseobacterium indologenes*. This is somewhat of a novelty. It is probably a good job in that they have done a lot of experiments, including making various mutants and measuring enzyme chemistry, and have examined CIM-1 closely. However, the finishing touches in the paper are so poor and crude. The data is inconsistent in some places, and I almost doubt that the content is even okay. I think that since the authors have done a lot of work, the paper would be much better if the authors paid more proper attention to detail.

I will list the points that I found problematic, but first of all, I have the impression that the results are very poorly presented in this paper. It could have been a very good paper with novel content, but the paper is so poorly finished that I might happen to doubt the content.

First of all, this may not be something to blame the authors for, but the text on L.217 is incomplete. It may be a system problem, but the beginning of line 217 does not connect to anywhere before it. However, this is not enough to obscure the entire paper, nor is it critical to change the impression.

The missing part from L217 was added (now 253 on page 13).

L.50

Roman B should be Greek beta.

This has been corrected (page 3, line 55)

L.128

Table Fig.1a does not match the MIC values in Table.1. Perhaps the MIC measurements performed in this study are those listed in Table.1, but what is the origin of the table in Fig.1a? There is no citation, and I cannot understand. Is it necessary?

We agreed with reviewer that Fig 1a is not necessary and it was removed.

L.172

The scale of the vertical axis is significantly different between Fig.2b and c. Is this a problem in the interpretation of the data in this study?

Figures 2b and c were removed in response to the comments from reviewer 1.

L.211

Fig.3d, there are lanes for Expression Level and Whole Cell lysate. What is the difference between them? I can't figure it out from reading the Materials and Methods. In some lanes, there is almost no expression (CIM-1 is a notable example), however the whole cell lysate and other fractions have protein bands, which is difficult to understand. Is it necessary to show both? How about reexamining the results, including the question of whether it is necessary to show both?

Thank you for your comment. The expression level was included to show that the protein expression levels vary between the different constructs and that the antimicrobial susceptibility correlates with the expression levels of the proteins. In order to avoid confusion, we have moved the western blot showing the expression levels to the Supplementary material (Supplementary Figure 4).

L.211

Furthermore, about this Fig. 3d,

A panel for CIM-1 Δ Sig is not shown, which I think is necessary. Also, there is a description in the Materials and Methods, so perhaps the authors simply forgot to include it?

We apologise for this oversight. The localisation of CIM-1 Δ Sig was accessed and included into Figure 3b.

Figure 3. Schematic representation of constructs and their postulated cellular localisation.

a) Schematic representation of the postulated cellular localization of the constructs. **b)** Representative Western blot images of protein levels of various protein constructs in whole cells and cellular fractions from *E. coli* C41(DE3). THE™ His tag monoclonal mouse antibodies and Invitrogen™ Goat anti-Mouse Immunoglobulin G was used in immunodetection.

L.227

The referring 3c on L.227, but my impression from what I have read is that this should be 3d?

The figure showing the expression level has been moved to the Supplementary material (Supplementary Figure 4) and is correctly referenced now (line 253 on page 14).

Expression level of all constructs in *E. coli* C41(DE3)

Supplementary Figure 4. Relative expression level of all His-tagged constructs (CIM-1, IND-2, NDM-4, CIM-1ΔSig, CIM-1 C19A, I-CIM, N-CIM, C-NDM and C-IND) in *E. coli* C41(DE3) cells. Cells were inoculated from an overnight culture and grown the same manner as preparation for antimicrobial susceptibility assay.

L.231 and another lines

There should be uniformity in the way proteins and mutants are described. L.231, it is CIM, but mostly CIM-1 are used. It would be more reader-friendly to unify the writing style. There is also another inconsistency on L.486 as well. In Supplementary Fig. 5, there are many. CIND / C-IND, ICIM / I-CIM. They should be fixed. Efforts should be made to unify this as it makes it difficult to understand. Even scientific content might be underestimated.

L 231 was removed from manuscript (now line 285 page 14). The way the proteins and mutants are described has been standardised throughout the manuscript.

L.233

Also, 3b should be 3d?

This has been corrected.

L.306.

I think it's fair to say that CIM-1 is identified, but I don't feel that IND-2 is identified in this paper. Isn't it enough that this paper has one significant study of CIM-1, as the title says? Or is there not much more to IND-2 than the Nordmann et al. paper in 2000? If there is an argument you want to emphasize, you should add a little more to that section.

IND-2 has been identified before (Bellais, Poirel, Leotard, Naas, & Nordmann, 2000) and hence was not the topic of this study. In this study we used IND-2 as a comparison of a soluble MBL in *C. indologenes*.

The manuscript has been amended (line 394 and 398, page 20).

"In this study, a putative MBL (CIM-1) was identified in a C. indologenes isolate recovered from a healthcare facility."

L.353

The OMV appears for the first time in this paper as an abbreviation, so it should not be abbreviated here.

Thank you for your comment. OMV is now first mentioned in the introduction (page 3, line 79) and was spelled out for the first time.

L457

The authors described in the Materials and Methods that BC21(DE3) is used in this study. Where is this cell used? Looking at Table.2, it seems that it is all expressed in C41(DE3).

E. coli BL21(DE3) was removed from the methods of the section on the antimicrobial susceptibility assay (line 569, page 27).

E. coli BL21(DE3) was used in protein expression and purification (line 545, page 25).

L.482

There is a difference between the mutants listed in L486 and those listed in Fig3. Wouldn't they be the same?

We apologise for this oversight and the Materials and methods section was amended (line 602-603, page 28).

"E. coli C41(DE3) expressing CIM-1, IND-2, NDM-4, CIM-1ΔSig, CIM-1 C19A, I-CIM, N-CIM, C-NDM and C-IND proteins were inoculated from an overnight culture and grown in 50 mL LB broth at 37 °C until the cells reached OD600 0.4-0.6."

Supplementary data L.45

Isn't CIM-1 in Supplemental Fig. 3 also CIM-1ΔSig?

We apologise for this oversight. CIM-1 in Supplementary Figure 3 (Now Supplementary Figure 10) was amended to CIM-1ΔSig.

Supplementary data L.45

Supplemental Fig. 3 shows the actual curves for the measurements summarized in Table-2, but why is only this one selected to be shown? Couldn't you also show data for other drugs or other series of mutants? It may be too much to show one by one, but it could be organized well by overlaying the three proteins (including mutants) for each drug with different markings (▲, etc.).

Supplementary Fig. 3 (now Fig. S10) does not show the actual curves for the measurements summarized in Table 2. It shows the inhibition effect of zinc specific metal chelator to our newly identified MBL, CIM-1 Δ Sig As IND and NDM were characterised MBLs, the inhibition assay was only done on CIM-1 Δ Sig.

Supplementary data L.67

L.67 of the supplementary information, they have the descriptions as CIM-1, CIM-1 C19A just above the two gel pictures, but the way it is written from line 68, shouldn't it be written on the bottom edge of the gel? And instead, there are no explanations for each lane, so I cannot understand what is being loaded.

This figure was removed from this paper in response to the comments from Reviewer 1.

Supplementary data L.69

Why is there a solid band on the Empty lane? It appears more intense than the band of Expression levels of CIM-1 in Fig. 3d. Is this paper okay? I'm just wondering.

I think that since you have done a lot of work, the paper would be much better if you paid more proper attention to detail.

Also, CIM-1 C19A on the right side of line 67 is the same figure as the left half of CIM-1 C19A on line 68, second from the left (in that sense, it seems like a useless duplication). However, the CIM-1 on the left side of line 67 is different from the CIM-1 on the left side of line 68. Why are these twice the same but different? I have a hard time understanding this. I may be going to be suspicious of this paper itself. Is it a simple mistake?

Thank you for your comment. The band observed on the empty lane was a leakage while loading the sample, thus the lane was skipped. The Western blot showing expression levels of different constructs was now moved to Supplementary Figure 4.

The membrane fractionation experiment was repeated, and all uncut gels were included in the Supplementary Figure 12.

My conclusion is that I believe that this work has sufficient potential to deserve recognition, but the paper is apparently poor. Please, to value the effort, time and resources spent, take care of the presentation of the data.

All the figures have been updated and improved.

Reviewer #3 (Remarks to the Author):

The work of Wang et al. presents the identification and characterization of a lipidated metallo- β -lactamases, CIM-1, from the opportunistic pathogen *Chryseobacterium indologenes*. CIM-1 is the second reported lipidated MBL and possesses an atypical lipobox that could potentially influence its expression, localization and activity. Overall, this study provides valuable insights into the characterization of CIM-1. However, to enhance the impact and broaden the interest of this work, it could benefit from additional functional characterization of the atypical lipobox and further evidence supporting the potential evolutionary advantages associated with this unique sequence.

Major points:

1. The authors demonstrated the hydrophobic interaction between CIM-1 and the membrane by showing its solubilization with Triton X-100. However, it remains unclear whether CIM-1 associates with the inner membrane or the inner leaflet of the outer membrane. Considering the authors' speculation regarding CIM-1 secretion in OMVs, it is crucial to confirm whether CIM-1 is associated with the outer membrane. Additionally, does CIM-1 possess any signal to avoid the LolCDE translocation system?

Thank you for your comment. In response to the comments from Reviewer 1, membrane solubilisation was removed and replaced with mass spectrometric analysis of *C. indologenes* membrane fractions and outer membrane vesicles. We predict that CIM-1 is attached to the outer membrane as it does not have the LolCDE avoidance signal (Asp at +2 position). The current data is not able to distinguish between inner/outer membrane localisation. However, we were able to show that CIM-1 predominantly associates with the membrane compared to the soluble protein IND-2 in the native host *C. indologenes*. Section 4 and Figure 4 was added to the manuscript to reflect these results.

Figure 4. CIM-1 is identified in the membrane fraction of *C. indologenes*.

a) Relative protein abundance of CIM-1 and IND-2 presented in the membrane fraction of *C. indologenes* #3362 normalised by protein expression level. b) Ratio of protein presented in membrane and outer membrane vesicles isolated from *C. indologenes* #3362

2. There appears to be missing content before lane 217. Moreover, the “expression levels” presented in Fig 3.d are somewhat perplexing. Are these western blots performed using antibodies against any specific tags? If not, comparing the expression levels of different lactamases could be challenging due to potential differences in antibody efficiency. Furthermore, the “soluble fraction” of N-CIM exhibits a smaller molecular weight than the band in the membrane fraction. Also, two bands can be seen in the soluble fraction of I-CIM. Could these bands be a result of proteolysis of the membrane-bound CIM?

Thank you for your comment. The missing content before lane 217 was added (now 253 on page 13).

Yes, same antibodies (anti-His antibodies) were used in comparing the expression levels of different beta-lactamases. The Method section was updated to reflect this (line 610-616, page 28).

Two bands can be seen in the both soluble and membrane fraction of I-CIM, however, the major band in the soluble fraction has a smaller size comparing to membrane fraction. We believe that the smaller protein in the soluble fraction is the mature (folded) protein of I-CIM. For the bigger sized protein in membrane fraction, it is potentially the precursor (full-length) protein which still haven't been processed by signal peptidase I. As I-CIM is so highly expressed, the precursor protein can be stuck during the process of signal peptidase cleavage therefore has some interaction with membranes. Regarding N-CIM, most of the protein is localized in the membrane fraction. The bigger sized N-CIM could potentially be the precursor (unfolded) protein.

3. The authors proposed that CIM-1 C19A associates with membrane through electrostatic interactions. It would be valuable to investigate the surface properties of CIM-1 based on the structural model. Does CIM-1 possess a surface area with positive electrostatic potential that might facilitate its membrane association?

Modelling of CIM-1 reveals a positive surface charge that could potentially facilitate membrane association. However, in accordance with the comments from Reviewer 1, we have removed Figure 2 and replaced this with data on the behaviour of CIM-1 in the native host, *C. indologenes*. These data are presented in Section 4 including Figure 4 of the Results.

4. It is evident that the presence of the atypical lipobox sequence significantly affects the level of CIM-1 when recombinantly expressed in *E. coli*. It remains unclear whether the same effect is observed in *C. indologenes*. It would be intriguing to explore whether this atypical yet conserved lipobox sequence confers any evolutionary benefits. The authors raised the intriguing possibility that CIM-1 could be excreted into the media through OMVs. Is it feasible to experimentally test this hypothesis and compare the secretion efficiency of CIM-1 with typical or atypical lipobox sequences?

Thank you for your comment. RT-qPCR shows that CIM-1 has a much lower expression level comparing to IND-2. (Supplementary Fig.8). Based on current experimental results, it is not clear whether this atypical lipobox confers any evolutionary benefit.

Our results shows that CIM-1 can be excreted in OMVs (Supplementary Fig. 7 and 9). Based on the high affinity of CIM-1 and frequent isolation of *C. indologenes* in co-infections, CIM-1 could potentially give population resistance.

It will be interesting to compare the secretion efficiency of CIM-1 with typical or atypical lipobox sequences in *C. indologenes*. However, there is no available literature about genetic manipulation of *C. indologenes*, it is very difficult to perform this experiment in the timeframe as the tools for genetic manipulation of *C. indologenes* must be established first. We also looked for the gene manipulation method for closely relative bacteria such as *Flavobacterium* spp. and found that *recB* and *recC* are absent, and RecA dependent homologous recombination is rare in *Flavobacterium* (Staroscik et al., 2008). Unfortunately, *recB* and *recC* are also absent in *C. indologenes*. This work falls outside of the remit of this

manuscript but is the topic of another study following from the research reported in this manuscript.

5. Studies have indicated that membrane anchoring contributes to the stability of NDM-1 (Nat Chem Biol. 2016 Jul;12(7):516-22), could the atypical lipobox sequence play a similar role in CIM-1 stability?

This is an interesting possibility. However, this would be best studied in the native host to ensure correct folding and localisation of the protein. As there is no available literature about genetic manipulation of *C. indologenes*, it is very difficult to perform this experiment in the timeframe as the tools for genetic manipulation of *C. indologenes* must be established first. This work falls outside of the remit of this manuscript but is the topic of another study following from the research reported in this manuscript.

6. It is essential to indicate the number of replicates performed for the MIC and western blot experiments, as well as specify the antibodies used.

Material and methods were updated to indicate the above (line 575-578, page 27).

“All assays were repeated at least four times on four different days with different batches of cells.”

References:

- Bellais, S., Poirel, L., Leotard, S., Naas, T., & Nordmann, P. (2000). Genetic diversity of carbapenem-hydrolyzing metallo-beta-lactamases from *Chryseobacterium* (Flavobacterium) *indologenes*. *Antimicrob Agents Chemother*, 44(11), 3028-3034. doi:10.1128/AAC.44.11.3028-3034.2000
- Prunotto, A., Bahr, G., Gonzalez, L. J., Vila, A. J., & Dal Peraro, M. (2020). Molecular Bases of the Membrane Association Mechanism Potentiating Antibiotic Resistance by New Delhi Metallo-beta-lactamase 1. *ACS Infect Dis*, 6(10), 2719-2731. doi:10.1021/acinfecdis.0c00341
- Staroscik, A. M., Hunnicutt, D. W., Archibald, K. E., & Nelson, D. R. (2008). Development of methods for the genetic manipulation of *Flavobacterium columnare*. *BMC Microbiol*, 8, 115. doi:10.1186/1471-2180-8-115

REVIEWERS' COMMENTS:

Reviewer #2 (Remarks to the Author):

This is the second time this reviewer had a chance to review this manuscript. The authors improved figures to substantiate their findings from their studies. It is commendable that the author's arguments were strengthened by the addition of new figures based on advice from other reviewers. And the authors have addressed the previous criticism constructively, and the revised manuscript is significantly improved. I do believe that this revised manuscript provides new insights into a new metallo-beta-lactamase (MBL) from *Chryseobacterium* (named CIM-1), which are worth sharing with Communications Biology readers.

Reviewer #3 (Remarks to the Author):

The authors have addressed all of my concerns. The only further change I can suggest is that the formulation in line 227-228 is not very logical, so I would suggest the following change: "..., suggesting that CIM-1 may be able to be translocated to the outer membrane via the Lol pathway, so we speculate that it may be located in the inner leaflet of the outer membrane". Other than that, I am happy to see this manuscript accepted for publication.